# Anterior insular cortex plays a critical role in interoceptive attention

Xingchao Wang[1,2†], Qiong Wu[3,4†], Laura Egan[5], Xiaosi Gu[6,7,8], Pinan Liu[1,2], Hong Gu[9], Yihong Yang[9], Jing Luo[3], Yanhong Wu[4,10]*, Zhixian Gao[1,2]*, Jin Fan[5,6,7,11]*

[1]Department of Neurosurgery, Beijing Tiantan Hospital, Capital Medical University, Beijing, China; [2]China National Clinical Research Center for Neurological Diseases, Beijing, China; [3]Beijing Key Lab of Learning and Cognition, School of Psychology, Capital Normal University, Beijing, China; [4]School of Psychological and Cognitive Sciences, Peking University, Beijing, China; [5]Department of Psychology, Queens College, The City University of New York, New York, United States; [6]Department of Psychiatry, Icahn School of Medicine at Mount Sinai, New York, United States; [7]Nash Family Department of Neuroscience, Icahn School of Medicine at Mount Sinai, New York, United States; [8]The Mental Illness Research, Education, and Clinical Center, The James J. Peter Veterans Affairs Medical Center, New York, United States; [9]Neuroimaging Research Branch, Intramural Research Program, National Institute on Drug Abuse, Baltimore, United States; [10]Beijing Key Laboratory of Behavior and Mental Health, Peking University, Beijing, China; [11]Friedman Brain Institute, Icahn School of Medicine at Mount Sinai, New York, United States

*For correspondence:
wuyh@pku.edu.cn (YW);
gaozx@ccmu.edu.cn (ZG);
jin.fan@qc.cuny.edu (JF)

†These authors contributed equally to this work

Competing interests: The authors declare that no competing interests exist.

**Abstract** Accumulating evidence indicates that the anterior insular cortex (AIC) mediates interoceptive attention which refers to attention towards physiological signals arising from the body. However, the necessity of the AIC in this process has not been demonstrated. Using a novel task that directs attention toward breathing rhythm, we assessed the involvement of the AIC in interoceptive attention in healthy participants using functional magnetic resonance imaging and examined the necessity of the AIC in interoceptive attention in patients with AIC lesions. Results showed that interoceptive attention was associated with increased AIC activation, as well as enhanced coupling between the AIC and somatosensory areas along with reduced coupling between the AIC and visual sensory areas. In addition, AIC activation was predictive of individual differences in interoceptive accuracy. Importantly, AIC lesion patients showed disrupted interoceptive discrimination accuracy and sensitivity. These results provide compelling evidence that the AIC plays a critical role in interoceptive attention.

DOI: https://doi.org/10.7554/eLife.42265.001

## Introduction

Our brain consistently receives physiological signals arising from sensory inputs and our body. Although attention toward inputs from the external environment (i.e., exteroceptive attention) has been extensively investigated, the attentional mechanism of the awareness and conscious focus on bodily somatic and visceral signals or responses (i.e., interoceptive attention) has been less studied because of difficulties in its measurement (*Brener and Ring, 2016*; *Ring et al., 2015*; *Craig, 2002*; *Craig, 2003*; *Craig, 2010*; *Critchley, 2005*; *Critchley, 2004*; *Farb et al., 2013a*). Previous theories argue that subjective emotions arise from these bodily reactions and visceral experiences

(*Cannon, 1987*; *Critchley and Harrison, 2013*; *Damasio, 1996*; *Dolan, 2002*; *Tranel and Damasio, 1991*) in which interoceptive awareness plays a critical role. Appropriate attention to bodily states and accurate perception of interoceptive information are essential in emotional awareness and in the maintenance of normal physiological conditions (*Craig, 2002*; *Craig, 2003*; *Craig, 2010*; *Critchley, 2005*; *Wiens, 2005*). The link between deficits in interoceptive attention and psychiatric symptoms may also be explained by the James–Lange theory of emotion (*Cannon, 1987*), the somatic marker hypothesis (*Damasio, 1996*; *Damasio et al., 1991*) for the embodied mind mediated by interoception (*Garfinkel and Critchley, 2013*), and the embodied predictive processing model (*Allen et al., 2016*; *Allen and Friston, 2018*; *Barrett and Simmons, 2015*; *Seth, 2013*; *Seth and Critchley, 2013*; *Seth et al., 2011*).

Recent human studies have emphasized the role of the insula in interoceptive representations (*Daubenmier et al., 2013*; *Farb et al., 2013b*; *Ronchi et al., 2015*). Neuroanatomical evidence, consistent with neuroimaging findings, suggests that the anterior insular cortex (AIC) is an important structure for encoding and representing interoceptive information (*Craig, 2002*; *Craig, 2003*; *Craig, 2009*; *Critchley et al., 2004*; *Stephani et al., 2011*). Although the AIC has been recognized as an interoceptive cortex (*Craig, 2003*; *Critchley et al., 2004*; *Ernst et al., 2014*; *Singer et al., 2009*; *Terasawa et al., 2013*), these findings remain equivocal because AIC activation seems ubiquitous across a wide range of tasks involving cognition, emotion, and other cognitive processes in addition to interoceptive attention (*Allen et al., 2013*; *Seeley et al., 2007*; *Uddin et al., 2014*; *Yarkoni et al., 2011*). Therefore, a task that selectively and reliably engages interoceptive attention needs to be employed. In addition, the correlational AIC activation found in functional neuroimaging studies alone does not provide causal evidence for its role in interoceptive attention, leaving the question of whether the AIC is critical in interoceptive attention unanswered. Studying patients with focal lesions in the AIC (*Gu et al., 2012*; *Gu et al., 2015*; *Ronchi et al., 2015*; *Starr et al., 2009*; *Wang et al., 2014*; *Wu et al., 2019*) would thus provide a unique opportunity to examine the necessity of the AIC in this fundamental process.

One challenge to the study of interoceptive attention is the vague nature of interoceptive awareness. According to the classic definition of attention by *James (1890)*, only the contents that are clearly perceived and represented by the mind can be the target of attention. However, most existing tasks measuring interoceptive attention fail to meet this criterion (*Ring et al., 2015*). In contrast to exteroceptive attention toward external sensory inputs, precise measurements of interoceptive attention are difficult to obtain experimentally because of the imprecise perception of visceral changes, such as heart rate (*Brener and Ring, 2016*; *Paulus and Stein, 2010*; *Ring et al., 2015*; *Windmann et al., 1999*). Multiple sources of physical information contribute to bodily signals, and most of these sources of somatic feedback cannot be described accurately by mindful introspection in normal physiological states (*Ring et al., 2015*). This limitation impedes accurate measurement of interoceptive attention and examination of the neural mechanisms underlying this process. To overcome this barrier, a perceivable visceral channel needs to be used.

Breathing is an essential activity for maintaining human life and, more importantly, is an easily perceivable bodily signal. As an autonomous vital movement, breathing can be measured and actively controlled in humans (*Daubenmier et al., 2013*; *Davenport et al., 2007*). The unique physiological characteristics of respiration render breath detection an ideal method for measuring interoceptive accuracy and sensitivity (*Garfinkel et al., 2015*) and for exploring the neural activity underlying interoceptive attention. Thus, we designed a breath detection task to engage interoceptive attention (attention to bodily signals), in which participants were required to indicate whether a presented breathing curve is delayed or not relative to their own breathing rhythm (breath detection task, BDT), in contrast to engaging exteroceptive attention (attention to visual signals), in which participants were required to indicate whether a visual dot stimulus is flashed on the breathing curve (dot flash detection task, DDT). This design enabled us to examine the involvement of the AIC in interoceptive processing in healthy participants and the necessity of the AIC in this processing in patients with AIC lesions.

Basing from previous evidence (e.g., *Critchley, 2004*), we hypothesized that the AIC is critical for interoceptive attention to reach subjective awareness by integrating information from an individual's homeostatic state and the external environment. We first conducted functional magnetic resonance imaging (fMRI) studies with two samples to map the neural substrates underlying interoceptive attention to internal bodily signals in contrast to exteroceptive attention to external visual signals in

healthy participants while they performed the tasks. We then investigated the necessity of the AIC in interoceptive attention by assessing interoceptive attention in patients with focal AIC lesions (AIC group) in comparison to brain-damaged controls (BDC group, patients with lesions in areas other than insular- or somatosensory-related cortices) and matched neurologically intact normal controls (NC group). We predicted that the AIC is involved in interoceptive attention and that patients with AIC lesions would show deficits in performance on the interoceptive but not exteroceptive attention task.

## Results

### Behavioral results of the fMRI studies

Performance accuracy (%) and discrimination sensitivity (*d′*) in the BDT were 82.1 ± 14.7% and 2.2 ± 1.1 (mean ± SD) for the first sample, and 74.9 ± 9.6% and 1.6 ± 0.6 (mean ± SD) for the second sample, respectively, which were significantly above the chance levels (50% and 0 for accuracy and *d′*, respectively; For the first sample: $t(43) = 14.51$, p<0.001, Cohen's $d = 2.18$ for accuracy and $t(43) = 13.09$, p<0.001, Cohen's $d = 2.0$ for *d′*, respectively; For the second sample: $t(27) = 13.77$, p<0.001, Cohen's $d = 2.59$ for accuracy and $t(27) = 12.89$, p<0.001, Cohen's $d = 2.67$ for d′, respectively), but lower than the DDT accuracy of 87.3 ± 9.8% and *d′* of 2.6 ± 0.8 for the first sample ($t(43) = -2.36$, p=0.02, Cohen's $d = 0.35$ and $t(43) = -2.31$, p=0.03, Cohen's $d = 0.35$, respectively) and accuracy of 80.9 ± 14.7% and *d′* of 2.2 ± 1.1 for the second sample ($t(27) = -1.83$, p=0.08, Cohen's $d = 0.35$ and $t(27) = -2.83$, p=0.009, Cohen's $d = 0.50$, respectively). Participants were slower in terms of reaction time (RT) (only for the first sample) and less biased in BDT than in the DDT (RT: $t(43) = 2.89$, p=0.006, Cohen's $d = 0.44$ for the first sample, and $t(27) = 0.6$, p=0.55, Cohen's $d = 0.12$ for the second sample; β: $t(43) = -2.62$, p=0.01, Cohen's $d = 0.39$ for the first sample, and $t(27) = -4.32$, p<0.001, Cohen's $d = 0.80$ for the second sample) (see *Figure 1—figure supplements 1* and *2* for details of the behavior results for the first and second samples, respectively, in accuracy, RT, *d′*, and β. Data were plotted in R using 'raincloud' script (*Allen et al., 2018a*; *Allen et al., 2018b*); See *Table 1* for the statistics of behavioral results for the first and second samples). The split-half reliability of the BDT and DDT were 0.86 and 0.85 for the first sample, and 0.68 and 0.89 for the second sample, respectively.

For the first sample, the relative interoceptive accuracy was negatively correlated with the subjectively scored difficulty of the BDT relative to the DDT (Pearson $r = -0.43$, corrected p=0.02, Bayes Factor (BF) = 10.38), but not significantly correlated with the 'awareness of bodily processes' subtest of the body perception questionnaire (BPQ) after correction for multiple comparisons (Pearson $r = 0.27$, corrected p=0.38, BF = 0.86). No significant correlations were observed between relative interoceptive accuracy and subjective emotion experiences, including trait positive affective experience (measured by Positive and Negative Affect Schedule, PANAS) (*Watson, 1988*) (Pearson $r = 0.31$, corrected p=0.20, BF = 1.38), anxiety (Pearson $r = -0.006$, p>0.9, BF = 0.19) or depression score (Pearson $r = -0.002$, p>0.9, BF = 0.19). For the second sample, however, we did not find significant correlations between relative interoceptive accuracy and scores of questionnaires (awareness of bodily processes: Pearson $r = -0.17$, corrected p>0.9, BF = 0.33; trait positive affective experience: Pearson $r = 0.12$, corrected p>0.9, BF = 0.27; anxiety: Pearson $r = 0.29$, corrected p=0.56, BF = 0.69; depression: Pearson $r = 0.03$, corrected p>0.9; note that we did not collect subjective rating of task difficulty in the second sample). In addition, we also calculated correlation coefficients between task performance and questionnaires by pooling the two samples (See *Table 2* for Pearson correlation strength and Bayesian tests between all behavioral measures in the first sample, the second sample, and across the two samples).

### Imaging results of the whole brain analysis of the first fMRI study

#### Main effects of interoceptive attention and feedback delay, and their interaction

The main effect of interoceptive attention, compared to exteroceptive attention (BDT vs. DDT), was associated with enhanced activity in the cognitive control network (*Fan, 2014*; *Wu et al., 2015*; *Xuan et al., 2016*), including the AIC, the dorsal anterior cingulate cortex (ACC) and the supplementary motor area (SMA), and the superior frontal and the parietal cortices (the frontal

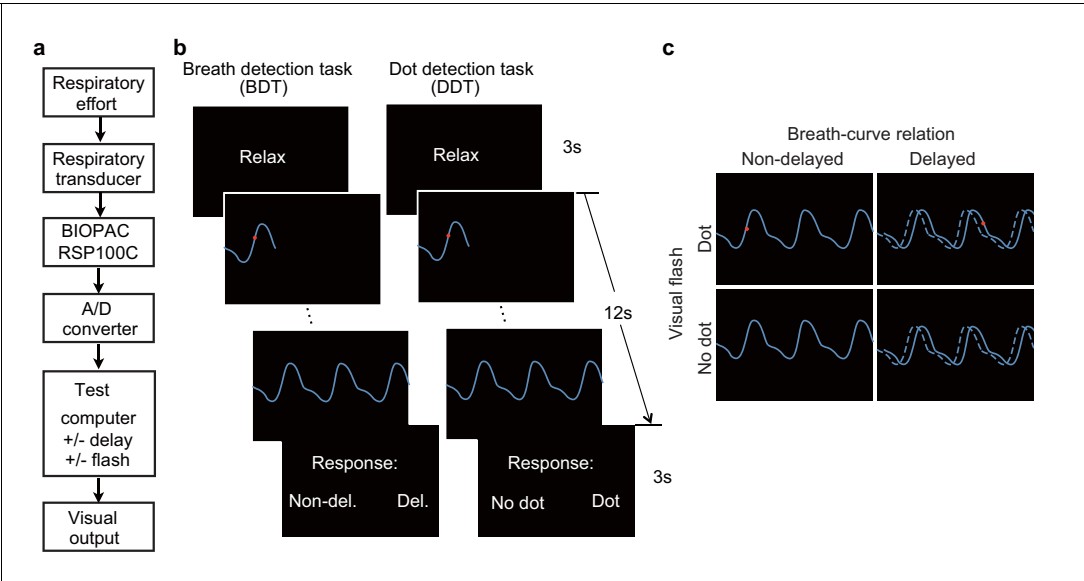

**Figure 1.** Experimental setup, trial structure of the tasks, and stimulus conditions. (**a**) The respiratory effort is converted into electronic signal changes using a respiratory transducer, amplified by BIOPAC, digitized using an A/D converter, and sent to the test computer for the final visual display as a dynamic breath curve, with or without a 400 ms delay. (**b**) This panel shows two trials for the breath detection task (BDT) and flash dot detection task (DDT) runs, respectively. Each trial begins with a 3 s blank display, followed by a 12 s display of respiratory curve presented with or without a 400 ms delay and with or without a 30 ms red dot flashed at a random position on the curve, and ends with a 3 s response window during which participants make a forced-choice button-press response to two alternative choices depending on the block type (BDT or DDT) to indicate whether the feedback curve is synchronous or delayed (for the BDT run) or whether a dot has appeared (for the DDT run). (**c**) The task represents a 2 × 2 × 2 factorial design with the factors of attention to breath or dot (block design), presence or absence of breath curve delay, and presence or absence of a dot flashed. The dashed line represents the actual breath curve, while the solid line represents the feedback breath curve displayed on the screen.

DOI: https://doi.org/10.7554/eLife.42265.002

The following source data and figure supplements are available for figure 1:

**Figure supplement 1.** Raincloud plots visualizing the five-number summary (minimum, lower quartile, median, upper quartile, and maximum) for (a) accuracy, (b) reaction time, (c) d', and (d) β for the BDT and DDT tasks in the first sample of the fMRI study.

DOI: https://doi.org/10.7554/eLife.42265.003

**Figure supplement 1—source data 1.** Behavioral data for the first sample of the fMRI study.

DOI: https://doi.org/10.7554/eLife.42265.004

**Figure supplement 2.** Raincloud plots visualizing the five-number summary (minimum, lower quartile, median, upper quartile, and maximum) for (a) accuracy, (b) reaction time, (c) d', and (d) β for the BDT and DDT tasks in the second sample of the fMRI study.

DOI: https://doi.org/10.7554/eLife.42265.005

**Figure supplement 2—source data 1.** Behavioral data for the second sample of the fMRI study.

DOI: https://doi.org/10.7554/eLife.42265.006

eye field, FEF; and the areas near/along the intraparietal sulcus; *Figure 2a*, *Table 3*). In addition, this contrast revealed significantly less activation, or deactivation, in the core regions of the default mode network (*Raichle et al., 2001*), including the the ventral medial prefrontal cortex, the middle temporal gyrus (MTG), and the posterior cingulate cortex.

Activation in the AIC, the middle frontal gyrus, the SMA, and the temporal parietal junction was associated with the main effect of feedback delay (*Figure 2b*, *Table 4*). The regions showing the main effect of feedback delay also showed the interaction effect between attentional focus (interoceptively in the BDT and exteroceptively in the DDT) and feedback (with and without delay) (*Figure 2c*, *Table 5*). The task-induced responses extracted from the bilateral AIC, defined by the attention by feedback interaction map, shows the activation pattern under different task conditions (*Figure 2d*). In specific, (1) the bilateral AIC demonstrated greater activation during interoceptive processing (BDT) than during exteroceptive processing (DDT), irrespective of the feedback type; (2) the delayed trials induced greater activation in the bilateral AIC in comparison to the non-delayed trials only during interoceptive processing (BDT). The evidence of this interaction effect in the AIC suggests that the AIC was actively engaged in interoceptive processing.

**Table 1.** Statistics of behavioral results of the fMRI studies.

| | | First sample | | | Second sample | | |
|---|---|---|---|---|---|---|---|
| | | Df | T | Cohen's d | Df | T | Cohen's d |
| accuracy | intero vs. 0.5 | 43 | 14.51*** | 2.18 | 27 | 13.77*** | 2.59 |
| | intero vs. extero | 43 | −2.36* | 0.35 | 27 | −1.83 | 0.35 |
| d' | intero vs. 0 | 43 | 13.09*** | 2.0 | 27 | 12.89*** | 2.67 |
| | intero vs. extero | 43 | -2.31* | 0.35 | 27 | -2.83** | 0.50 |
| β | intero vs. extero | 43 | −2.31* | 0.35 | 27 | −2.83** | 0.50 |
| RT | intero vs. extero | 43 | 2.89** | 0.44 | 27 | 0.6 | 0.12 |

* $p<0.05$; ** $p<0.01$; *** $p<0.001$.
DOI: https://doi.org/10.7554/eLife.42265.008

**Table 2.** Pearson correlation coefficients (and Bayes Factors) between the behavioral measurements for the first, the second, and across the two samples.

| | | Relative accuracy | Subjective difficulty | BPQ | Positive PANAS | HAMA | BDI |
|---|---|---|---|---|---|---|---|
| **1st sample** | Relative accuracy | - | | | | | |
| | Subjective difficulty | −0.43** (10.38) | - | | | | |
| | BPQ | 0.27 (0.17) | −0.15 (0.29) | - | | | |
| | Positive PANAS | 0.31 (1.38) | −0.04 (0.19) | −0.006 (0.19) | - | | |
| | HAMA | −0.006 (0.19) | −0.14 (0.28) | 0.25 (0.69) | −0.12 (0.25) | - | |
| | BDI | −0.002 (0.19) | −0.004 (0.19) | 0.16 (0.32) | −0.06 (0.20) | 0.70*** (>100) | - |
| **2nd sample** | Relative accuracy | - | | | | | |
| | Subjective difficulty | - | - | | | | |
| | BPQ | −0.17 (0.33) | - | - | | | |
| | Positive PANAS | 0.12 (0.27) | - | 0.07 (0.25) | - | | |
| | HAMA | 0.29 (0.69) | - | 0.40 (1.90) | −0.034 (0.24) | - | |
| | BDI | 0.034 (0.24) | - | 0.075 (0.25) | −0.43 (2.84) | 0.47* (4.96) | - |
| **1st + 2nd samples** | Relative accuracy | - | | | | | |
| | Subjective difficulty | - | - | | | | |
| | BPQ | 0.06 (0.17) | - | - | | | |
| | Positive PANAS | 0.25 (1.16) | - | 0.03 (0.15) | - | | |
| | HAMA | 0.12 (0.25) | - | 0.31* (4.91) | −0.09 (0.20) | - | |
| | BDI | 0.008 (0.15) | - | 0.14 (0.28) | −0.20 (0.56) | 0.60*** (>100) | - |

* corrected $p<0.05$; ** corrected $p<0.01$; *** corrected $p<0.001$; value in brackets represents Bayes factor. BPQ, body perception questionnaire; PANAS, positive and negative affective schedule; HAMA, Hamilton anxiety scale; BDI, Beck depression inventory.
DOI: https://doi.org/10.7554/eLife.42265.009

## Correlation between interoceptive accuracy and AIC activation

Voxel-wise regression analysis revealed the relationship between the interoceptive task-induced activation strength (map of the interaction contrast) and participants' interoceptive accuracy (performance accuracy in the BDT), with exteroceptive accuracy (performance accuracy in the DDT) controlled as a covariate. Higher interoceptive accuracy was associated with greater interaction effect of the bilateral AIC (and MTG) across participants (*Figure 3a*, *Table 6*). The AIC activation during the interoceptive processing involved attending to physiological signals and matching bodily signals to external visual input, which predicted individual differences in interoceptive attention (see *Figure 3b* for the illustration).

## Functional and effective connectivity of the AIC

Psychophysiological interaction (PPI) analysis showed augmented connectivity between the right AIC (as the seed) and the SMA/ACC, the FEF, the inferior frontal gyrus (IFG), and the postcentral gyrus (PoCG) during interoceptive (versus exteroceptive) attention (BDT vs. DDT) in contrast to the reduced connectivity between the right AIC and visual cortices (VCs) modulated by interoceptive attention (*Figure 4a*, *Table 7*). This result indicates that an increase in activation in the right AIC was associated with a greater increase in activation in the FEF, the IFG, and the PoCG and a greater decrease in activation in the VCs under interoceptive attention compared with exteroceptive

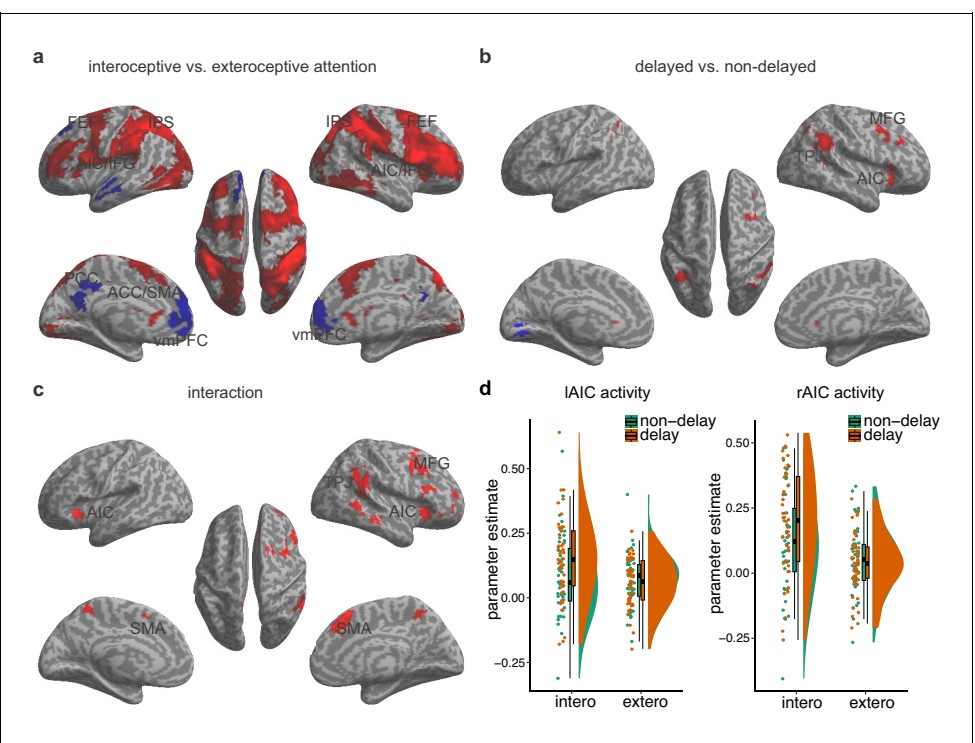

**Figure 2.** Main effects and the interaction effect of the whole brain analysis of the first sample. (a) Main effect of interoceptive vs. exteroceptive attention contrast (BDT vs. DDT). (b) Main effect of breath curve feedback condition (delayed curve vs. non-delayed curve). (c) Interaction between attention type and breath-curve feedback condition ([delayed – non-delayed] $_{BDT}$ – [delayed – non-delayed] $_{DDT}$). Here we showed the left AIC for the visualization of the seed for the ROI analysis in the second fMRI sample, although the cluster with 210 voxels did not survive GRF correction. Red color represents an increased activation; Blue color represents a decreased activation. (d) Activation of the left and the right AIC under the four task conditions, and the pattern of the interaction.

DOI: https://doi.org/10.7554/eLife.42265.010

The following source data is available for figure 2:

**Source data 1.** CSV file containing data for *Figure 2d*.

DOI: https://doi.org/10.7554/eLife.42265.011

**Table 3.** Activation and deactivation of the brain regions involved in interoceptive attention (interoception – exteroception).

| Region | L/R | BA | MNI X | Y | Z | T | Z | K |
|---|---|---|---|---|---|---|---|---|
| Positive | | | | | | | | |
| Cerebelum crus I | L | | −30 | −70 | −24 | 13.02 | Inf. | 73834 |
| Middle occipital gyrus | R | 19 | 32 | −68 | 22 | 11.99 | Inf. | |
| Cerebelum crus II | L | | −20 | −78 | −48 | 11.72 | 7.80 | |
| Inferior frontal gyrus | R | 44 | 52 | 14 | 24 | 11.24 | 7.63 | |
| Inferior parietal lobule | R | 40 | 36 | −48 | 44 | 11.19 | 7.62 | |
| Inferior parietal lobule | L | 40 | −38 | −46 | 42 | 10.41 | 7.32 | |
| Postcentral gyrus | R | 2 | 46 | −40 | 54 | 10.29 | 7.27 | |
| Supramarginal gyrus | R | 40 | 48 | −34 | 42 | 10.00 | 7.15 | |
| Superior occipital gyrus | R | 7 | 22 | −72 | 46 | 9.99 | 7.15 | |
| Cerebelum VIIB | L | | −32 | −70 | −52 | 9.78 | 7.06 | |
| Superior parietal lobule (Intraparietal sulcus) | R | 7 | 16 | −78 | 52 | 9.69 | 7.02 | |
| Cerebelum VIII | R | | 22 | −74 | −50 | 9.61 | 6.99 | |
| Middle frontal gyrus | L | 46 | −44 | 50 | 12 | 9.20 | 6.80 | |
| Middle frontal gyrus | R | 46 | 42 | 42 | 24 | 9.16 | 6.78 | |
| Supplementary motor area | R | 6 | 8 | 4 | 76 | 8.92 | 6.68 | |
| Inferior occipital gyrus | R | 37 | 52 | −66 | −12 | 8.68 | 6.56 | |
| Cerebelum crus II | R | | 2 | −76 | −36 | 8.66 | 6.56 | |
| Middle occipital gyrus (Intraparietal sulcus) | R | 19 | 32 | −76 | 34 | 8.58 | 6.52 | |
| Thalamus | R | | 18 | −20 | 20 | 8.55 | 6.50 | |
| Inferior temporal gyrus | R | 20 | 56 | −38 | −20 | 8.41 | 6.43 | |
| Inferior frontal gyrus | R | 45 | 44 | 38 | 12 | 8.31 | 6.38 | |
| Superior parietal lobule (Intraparietal sulcus) | L | 7 | −20 | −72 | 46 | 8.21 | 6.33 | |
| Supplementary motor area | L | 6 | -2 | -4 | 74 | 8.08 | 6.27 | |
| Inferior frontal gyrus | L | 44 | −54 | 12 | 26 | 8.07 | 6.26 | |
| Caudate | R | | 16 | -8 | 24 | 7.89 | 6.17 | |
| Anterior cingulate cortex | R | 32 | 2 | 18 | 44 | 7.78 | 6.12 | |
| Vermis | | | -2 | −74 | −12 | 7.76 | 6.10 | |
| Middle frontal gyrus | R | 46 | 50 | 14 | 40 | 7.75 | 6.10 | |
| Middle frontal gyrus | L | 46 | −40 | 34 | 34 | 7.72 | 6.08 | |
| Supramarginal gyrus | L | 40 | −60 | −36 | 28 | 7.47 | 5.95 | |
| Middle frontal gyrus | R | 6 | 28 | 2 | 48 | 7.01 | 5.69 | |
| Anterior insular cortex | R | | 34 | 20 | 4 | 6.98 | 5.68 | |
| Postcentral gyrus | L | 2 | −62 | −26 | 36 | 6.87 | 5.62 | |
| Inferior frontal gyrus | L | 6 | −52 | 8 | 12 | 6.84 | 5.59 | |
| Superior frontal gyrus | L | 6 | −26 | 4 | 66 | 6.73 | 5.53 | |
| Middle occipital gyrus (Intraparietal sulcus) | L | 7 | −24 | −66 | 36 | 6.66 | 5.49 | |
| Lingual gyrus | L | 18 | −18 | −90 | −18 | 6.61 | 5.46 | |
| Superior parietal lobule | L | 1 | −24 | −44 | 72 | 6.55 | 5.42 | |
| Caudate | L | | -8 | 22 | 4 | 6.45 | 5.37 | |
| Precentral gyrus | L | 6 | −40 | 2 | 56 | 6.23 | 5.23 | |
| Superior occipital gyrus | L | 18 | −22 | −92 | 28 | 6.20 | 5.21 | |
| Middle occipital gyrus | L | 18 | −24 | −94 | 16 | 6.09 | 5.14 | |

*Table 3 continued on next page*

*Table 3 continued*

| Region | L/R | BA | MNI X | Y | Z | T | Z | K |
|---|---|---|---|---|---|---|---|---|
| Middle occipital gyrus | R | 18 | 30 | −86 | 16 | 6.09 | 5.14 | |
| Fusiform gyrus | L | 37 | −46 | −46 | −22 | 5.82 | 4.97 | |
| Anterior insular cortex | L | | −30 | 20 | 8 | 5.50 | 4.76 | |
| Cuneus | L | 19 | 0 | −88 | 34 | 5.22 | 4.57 | |
| Superior parietal lobule | L | 5 | −18 | −60 | 66 | 5.18 | 4.54 | |
| Fusiform gyrus | R | 37 | 44 | −32 | −20 | 4.96 | 4.39 | |
| Negative | | | | | | | | |
| Anterior cingulate cortex | R | 32 | 4 | 38 | -4 | 7.47 | 5.95 | 3232 |
| Anterior cingulate cortex | L | 32 | -6 | 38 | -4 | 7.10 | 5.94 | |
| Superior frontal gyrus | L | 9 | −16 | 38 | 54 | 5.97 | 5.07 | |
| Medial superior frontal gyrus | R | 32 | 10 | 52 | 20 | 5.33 | 4.65 | |
| Medial superior frontal gyrus | L | 32 | -8 | 50 | 26 | 5.32 | 4.63 | |
| Middle frontal gyrus | L | 8 | −24 | 30 | 56 | 5.12 | 4.50 | |
| Superior frontal gyrus | L | 9 | −20 | 32 | 48 | 4.54 | 4.08 | |
| Precuneus | L | 23 | −10 | −44 | 40 | 6.45 | 5.37 | 819 |
| Precuneus | R | 23 | 6 | −60 | 24 | 4.24 | 3.85 | |
| Middle temporal gyrus | L | 21 | −60 | −10 | −14 | 5.89 | 5.02 | 787 |

DOI: https://doi.org/10.7554/eLife.42265.012

attention (*Figure 4b*). Similar PPI results were obtained when the left AIC was used as the seed (*Figure 4—figure supplement 1*).

On the basis of the PPI results, VCs of the right V2/3 (x = 14, y = −90, z = 28 as indicated by negative PPI) and the right PoCG (x = 58, y = −16, z = 32 as indicated by positive PPI) were included in the dynamic causal modeling (DCM) model. Data from one participant were excluded because significant activation in the V2/3 region of interest could not be identified. For model comparison, random-effects (RFX) Bayesian model selection (BMS) indicated that the winning model (with an exceedance probability of 29.84%) was the one with the modulatory effects of interoceptive and

**Table 4.** Activation and deactivation of the brain regions involved in feedback delay (delay – non-delay).

| Region | L/R | BA | MNI X | Y | Z | T | Z | K |
|---|---|---|---|---|---|---|---|---|
| Positive | | | | | | | | |
| Anterior insular cortex | R | | 30 | 26 | -4 | 5.26 | 4.60 | 618 |
| Inferior frontal gyrus | R | 45 | 42 | 22 | 8 | 4.40 | 3.98 | |
| Caudate | R | | 8 | 24 | 4 | 4.29 | 3.90 | |
| Inferior parietal lobule | L | 40 | −38 | −54 | 42 | 5.23 | 4.58 | 598 |
| Angular gyrus | R | 39 | 44 | −44 | 30 | 4.99 | 4.41 | 1317 |
| Inferior parietal lobule | R | 40 | 56 | −54 | 44 | 4.17 | 3.80 | |
| Middle frontal gyrus | R | 6 | 34 | 8 | 46 | 4.78 | 4.26 | 780 |
| Middle frontal gyrus | R | 9 | 34 | 18 | 34 | 4.74 | 4.23 | |
| Middle frontal gyrus | R | 46 | 34 | 28 | 32 | 4.32 | 3.92 | |
| Negative | | | | | | | | |
| Lingual gyrus | L | 17 | −10 | −78 | -4 | 6.21 | 5.22 | 443 |

DOI: https://doi.org/10.7554/eLife.42265.013

**Table 5.** Activation of brain regions related to the interaction between interoceptive attention and feedback delay ([delayed – non-delayed] interoception – [delayed – non-delayed] exteroception).

| Region | L/R | BA | MNI X | Y | Z | T | Z | K |
|---|---|---|---|---|---|---|---|---|
| Positive | | | | | | | | |
| Anterior insular cortex | R | | 28 | 28 | 0 | 5.52 | 4.77 | 516 |
| Inferior frontal gyrus | R | 47 | 40 | 26 | −10 | 4.66 | 4.17 | |
| Middle frontal gyrus | R | 9 | 40 | 14 | 40 | 5.36 | 4.67 | 2330 |
| Supplementary motor area | R | 8 | 4 | 22 | 54 | 5.19 | 4.55 | |
| Anterior cingulate cortex | R | 32 | 6 | 36 | 38 | 5.12 | 4.5 | |
| Superior frontal gyrus | R | 8 | 6 | 30 | 44 | 4.71 | 4.21 | |
| Inferior frontal gyrus | R | 45 | 46 | 22 | 16 | 4.50 | 4.05 | |
| Middle frontal gyrus | R | 6 | 34 | 4 | 52 | 4.27 | 3.88 | |
| Supplementary motor area | L | 6 | −12 | 8 | 52 | 3.64 | 3.38 | |
| Anterior cingulate cortex | R | 32 | 10 | 30 | 28 | 3.49 | 3.25 | |
| Supramarginal gyrus | R | 40 | 54 | −46 | 26 | 4.91 | 4.35 | 1748 |
| Middle temporal gyrus | R | 21 | 66 | −32 | −10 | 4.70 | 4.20 | |
| Inferior parietal lobule | R | 19 | 60 | −48 | 42 | 4.56 | 4.10 | |
| Superior temporal gyrus | R | 42 | 58 | −40 | 16 | 4.49 | 4.04 | |

DOI: https://doi.org/10.7554/eLife.42265.014

exteroceptive attention (BDT and DDT) exerting on the connection from the AIC to the PoCG and from the AIC to V2/3 (*Figure 4c* and *Figure 4—figure supplement 2*). The BMS indicated that interoceptive and exteroceptive attention were achieved through modulating the top-down connectivity from the AIC to these two sensory cortices.

We performed parameter inference by using Bayesian model averaging (BMA), which considers uncertainty by pooling information across all models in a weighted fashion (*Stephan et al., 2010*). For BMA (*Figure 4d*), the modulatory effect of interoceptive attention (BDT) was significant on the connection from the AIC to the PoCG ($t(42)$ = 4.85, Bonferroni corrected p<0.001). The modulatory effect of exteroceptive attention (DDT) on the connection from the AIC to the V2/3 was significant without correction ($t(42)$ = 2.25, uncorrected p=0.03). The BMA results were consistent with the winning model selected by model comparison and the PPI results: the modulatory effect from the AIC to the PoCG was driven by interoceptive attention (BDT), whereas the modulatory effect from the AIC to the V2/3 was driven by exteroceptive attention (DDT). In addition, the BMA results highlighted the importance of the intrinsic efferent connection from the AIC to the PoCG in the network ($t(42)$ = 3.61, Bonferroni corrected p=0.01).

## Region-of-interest (ROI) analysis results of the second fMRI study

The interaction between attentional focus (interoceptively in BDT and exteroceptively in DDT) and feedback (with and without delay) was significant in both left and right AICs (left: $F(1, 27)$=6.12, p=0.020; right: $F(1,27)$ = 5.88, p=0.022; *Figure 5a*), which confirmed the interaction effect in the bilateral AIC revealed by whole brain analyses of the first sample. The main effect of attentional focus (BDT vs. DDT) was significant in the right AIC with greater activation during the BDT than during the DDT ($F(1, 27)$=4.20, p=0.05) but not significant in the left AIC ($F(1, 27)$<1, p=0.51). The main effect of the feedback was not significant in either left or right AIC (left: $F$<1; right: $F$<1). In addition, similar to the results of the first sample, we found a significant correlation between the interaction effect of both left and right AICs and relative interoceptive accuracy (left: Pearson $r$ = 0.32, p=0.050, one-tailed; right: Pearson $r$ = 0.42, p=0.014, one-tailed; *Figure 5b*). In addition, we examined the pattern of the respiratory volume under BDT and DDT (see *Figure 5—figure supplement 1*). Despite the difference in the respiratory volume between interoceptive and exteroceptive conditions (BDT and DDT was significant, $F(1,27)$ = 15.88, p<0.001), this difference was canceled out for the

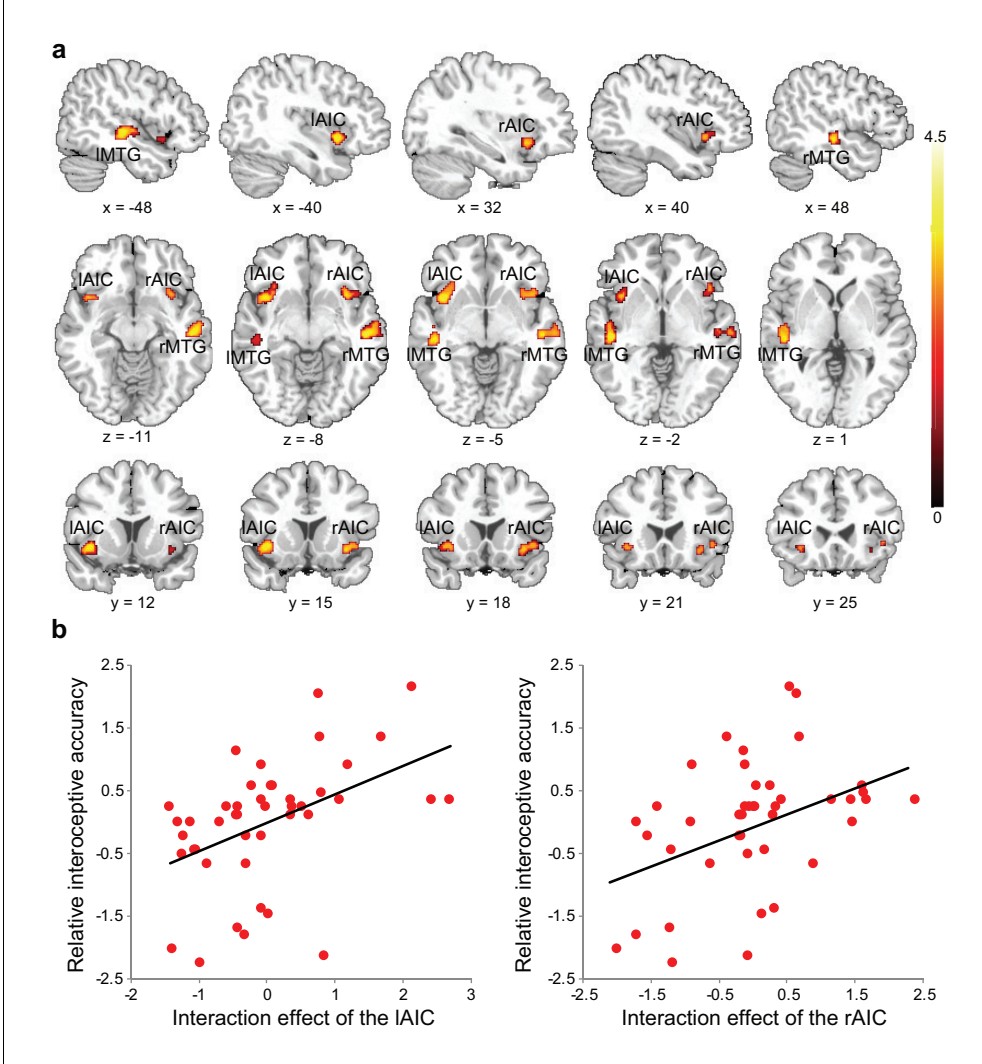

**Figure 3.** Relationship between brain activation and behavioral performance across participants. (a) This was revealed in a regression analysis of contrast images for the interaction between interoceptive attention deployment (BDT vs. DDT) and breath curve feedback condition (delayed vs. no-delayed), with performance accuracy on interoceptive and exteroceptive tasks as regressor-of-interest and covariate, respectively. AIC, anterior insular cortex; MTG, middle temporal gyrus. (b) Correlational patterns between the interaction effect of bilateral AIC activation and relative interoceptive accuracy. Data were normalized as z-scores.

DOI: https://doi.org/10.7554/eLife.42265.015

The following source data is available for figure 3:

**Source data 1.** CSV file containing data for *Figure 3b*.

DOI: https://doi.org/10.7554/eLife.42265.016

interaction effect ($F < 1$). These results further illustrated that the interaction effect in the AIC is not subject to the confounding of breathing effort difference between the two tasks.

The whole brain analysis of the second fMRI sample showed significant overlap between the activations without and with physiological correction for the main and the interaction effects (see *Figure 5—figure supplement 2*). We further checked how much physiological noise impacted AIC activation by comparing the contrast maps without and with physiological correction at an extremely permissive threshold ($p<0.05$ uncorrected). The difference in signals of the AIC between the analyses with and without physiological corrections was only evident for the main effect of interoceptive vs. exteroceptive attention (BDT vs. DDT) but not for the interaction contrast, confirming that the

**Table 6.** Relationship between the interaction effect ([delayed – non-delayed] interoception – [delayed – non-delayed] exteroception) of the brain and behavioral performance (interoceptive accuracy) across participants.

| Region | L/R | BA | MNI X | Y | Z | T | Z | K |
|---|---|---|---|---|---|---|---|---|
| Positive | | | | | | | | |
| Middle temporal gyrus | R | 20 | 54 | −20 | −10 | 3.85 | 3.53 | 232 |
| Middle temporal gyrus | L | 22 | −48 | −24 | -2 | 3.69 | 3.41 | 170 |
| Anterior insular cortex | L | | −42 | 12 | -6 | 3.64 | 3.37 | 168 |
| Anterior insular cortex | R | | 42 | 16 | -6 | 3.41 | 3.18 | 119 |
| Angular gyrus | R | 22 | 58 | −50 | 26 | 3.10 | 2.92 | 128 |

DOI: https://doi.org/10.7554/eLife.42265.017

interaction effect of the AIC was not significantly impacted by the physiological noises (see *Figure 5—figure supplement 3*). Altogether, these ROI results from the second sample confirmed that the AIC was actively engaged in interoceptive processing.

### Lesion study results: the necessity of the AIC in interoceptive attention

*Figure 6* shows the insular lesion overlap for the AIC patient group. The area with the most overlap was identified as the AIC according to the literature (*Kurth et al., 2010*; *Naidich et al., 2004*). We found a significant interaction effect between group (AIC, BDC, and NC) and task (BDT and DDT) in performance accuracy ($F_{(2,21)}$ = 5.19, p=0.015) and discrimination sensitivity ($d'$) ($F_{(2,21)}$ = 4.77, p=0.023). Planned simple comparisons were conducted between groups for each task. For the BDT, patients with AIC lesions had significantly lower performance accuracy (58%, $t$(13) = −3.47, p<0.001, BF = 14.71 compared with NC; $t$(8) = −2.35, p=0.009, BF = 3.95 compared with BDC) (*Figure 7a*) and discrimination sensitivity ($d'$) compared with the NCs and BDCs groups ($t$(13) = −3.62, p<0.001, BF = 13.78 compared with NC; $t$(8) = −2.22, p=0.013, BF = 3.40 compared with BDC) (*Figure 7b*), indicating diminished interoceptive attention. However, we did not find significant difference in accuracy between the NC and BDC groups ($t$(8) = 0, p=0.3; $d'$: $t$(8) = 0.112, p=0.23). For the DDT, the patients with AIC lesions did not show significant abnormalities in performance accuracy (AIC vs. NC: $t$(9) = 0.18, p=0.22, BF = 0.38; AIC vs. BDC: $t$(7) = −0.99, p=0.10, BF = 0.98), and in $d'$ (AIC vs. NC: $t$(9) = 0.18, p=0.22, BF = 0.38; AIC vs. BDC: $t$(7) = −0.83, p=0.12, BF = 0.85) compared with the NC and BDC groups. We did not find significant interaction effect on $\beta$ ($F$ <1, p=0.65) (*Figure 7d–f*). A summary of the statistical results of the lesion study is provided in *Table 8*. Our results demonstrated significant impairment in discrimination ability when attending to bodily signals, but not to external visual input, in patients with AIC lesions.

## Discussion

Using fMRI, we showed that the AIC is involved in interoceptive attention towards respiration, with the underlying connectivity between the AIC and the somatosensory cortex and visual areas modulated by interoceptive and exteroceptive attention, respectively. Notably, we confirmed the necessity of the AIC in supporting interoceptive attention by showing reduced behavioral performance on the interoceptive task in patients with focal AIC lesions. Thus, this study demonstrates that the AIC plays a critical role in interoceptive attention.

### The necessity of the AIC in interoceptive attention

Previous functional neuroimaging studies have shown that the insula is activated by autonomic arousal and emotional reactions (*Craig, 2002*; *Craig, 2003*; *Critchley et al., 2004*) and emphasized the central role of the insula in interoceptive awareness. The achievement of interoceptive awareness depends on the integration of afferent bodily signals with higher-order contextual information attributable to the AIC (*Craig, 2002*; *Craig, 2009*; *Critchley, 2005*; *Damasio et al., 2000*; *Mutschler et al., 2009*). In this study, the increase in neural activation in the AIC and other related brain structures when focusing on breath rhythm indicates that the AIC supports attention toward

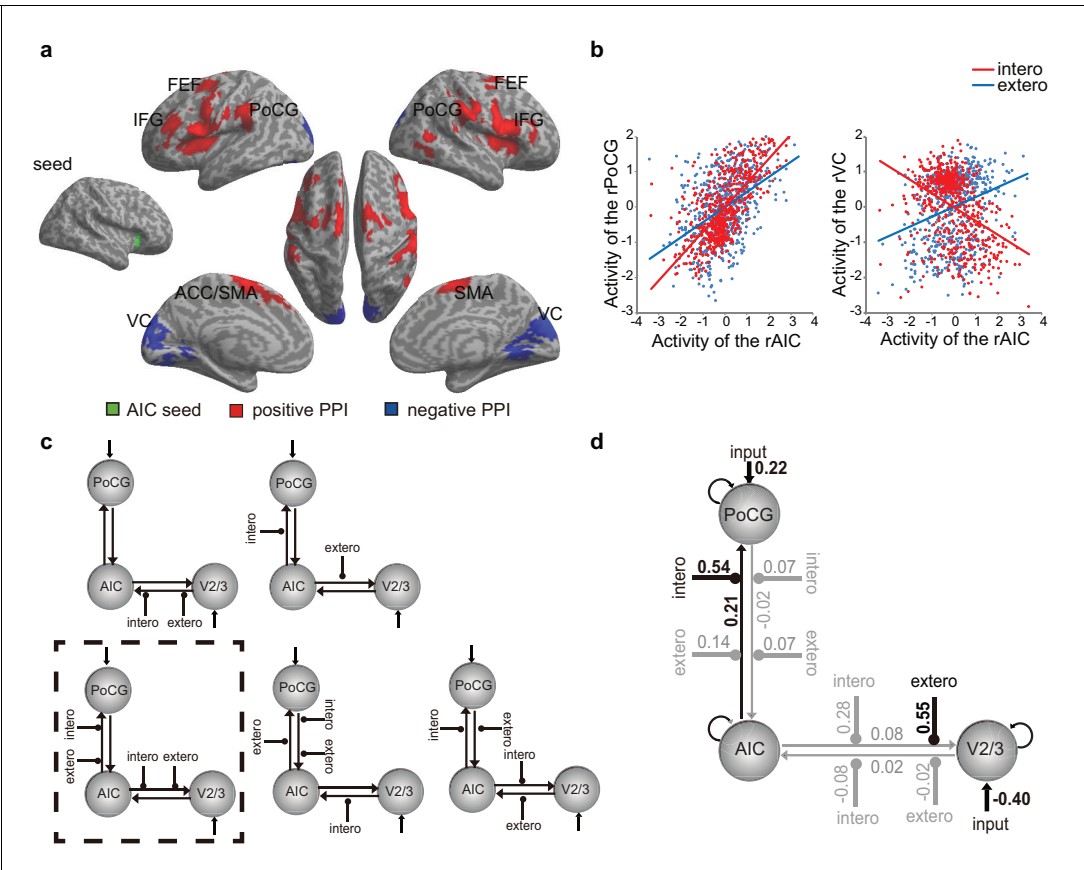

**Figure 4.** PPI and DCM results of the first fMRI sample. (**a**) Regions showing positive (red) and negative (blue) associations with AIC activation modulated by interoceptive attention relative to exteroceptive attention (BDT vs. DDT). (**b**) An increase in activation in the right AIC was associated with an increase in activation in the postcentral gyrus (PoCG) and a decrease in activation in the visual cortex (VC, V2/3) under the condition of interoceptive attention compared with exteroceptive attention. (**c**) Five base models generated by specifying possible modulations of interoceptive and exteroceptive attention (BDT and DDT) on the four endogenous connections between ROIs. The model surrounded by a rectangle in dashed-line indicates the winning model out of 52 variant models revealed by random-effects Bayesian model selection (BMS). (**d**) Intrinsic efferent connection from the AIC to the PoCG was significant. The modulatory effect of interoceptive attention (BDT) on the connection from the AIC to the PoCG was significant. The modulatory effect of exteroceptive attention (DDT) on the connection from AIC to V2/3 was significant (uncorrected).
DOI: https://doi.org/10.7554/eLife.42265.018

The following source data and figure supplements are available for figure 4:

**Source data 1.** CSV file containing data for *Figure 4b*.
DOI: https://doi.org/10.7554/eLife.42265.021
**Figure supplement 1.** Regions showed positive (red) and negative (blue) association with the left AIC (as the seed) modulated by interoceptive attention relative to exteroceptive attention (BDT vs DDT) for the first fMRI sample.
DOI: https://doi.org/10.7554/eLife.42265.019
**Figure supplement 2.** Exceedance probability of RFX BMS for the first fMRI sample.
DOI: https://doi.org/10.7554/eLife.42265.020

bodily signals. Most importantly, participants' performance accuracy on the interoceptive task was significantly correlated with the activation of the AIC, further demonstrating the involvement of the AIC in interoceptive attention.

Anatomically, the insula receives thalamo-insular projections of the interoceptive pathways (*Craig, 2002*). The AIC encodes subjective feelings (*Craig, 2003*; *Craig, 2009*; *Flynn, 1999*) and is critical for instantaneous representation of the state of the body (*Allen et al., 2016*; *Allen and Friston, 2018*; *Cao et al., 2014*; *Gu et al., 2015*). During the BDT, this is achieved by attending to bodily signals (i.e., breath rhythm) and matching them to external visual feedback (i.e., the breath curve). The present results provide additional support to previous finding that the activation of the

**Table 7.** Positive and negative psychophysiological interaction effects with the right AIC as the seed.

| Region | L/R | BA | MNI X | Y | Z | T | Z | K |
|---|---|---|---|---|---|---|---|---|
| Positive | | | | | | | | |
| Inferior frontal operculum | R | 44 | 52 | 8 | 26 | 7.49 | 5.96 | 5895 |
| Precentral gyrus | R | 6 | 58 | 10 | 36 | 6.71 | 5.52 | |
| Insula cortex | R | | 38 | 0 | 14 | 6.35 | 5.30 | |
| Putamen | R | | 20 | 8 | 10 | 6.33 | 5.29 | |
| Rolandic operculum | R | 48 | 48 | 4 | 10 | 6.01 | 5.09 | |
| Caudate | R | | 8 | 10 | 4 | 5.86 | 5.00 | |
| Inferior frontal gyrus | R | 45 | 42 | 36 | 10 | 4.35 | 3.94 | |
| Postcentral gyrus | R | 43 | 58 | −16 | 32 | 6.95 | 6.55 | 2078 |
| Supramarginal gyrus | R | 2 | 66 | −22 | 34 | 6.04 | 5.11 | |
| Superior temporal gyrus | R | 42 | 62 | −32 | 20 | 5.28 | 4.61 | |
| Precentral gyrus | L | 6 | −58 | 10 | 30 | 6.89 | 5.63 | 11155 |
| Putamen | L | | −20 | 10 | 12 | 6.04 | 5.11 | |
| Supplementary motor area | L | 6 | -8 | -4 | 64 | 5.90 | 5.02 | |
| Caudate | L | | -8 | 16 | 2 | 5.41 | 4.70 | |
| Triangle Inferior fronal gyrus | L | 48 | −38 | 32 | 24 | 5.21 | 4.56 | |
| Superior temporal gyrus | L | 44 | −48 | −42 | 24 | 5.19 | 4.55 | |
| Insula cortex | L | | −36 | -2 | 8 | 5.19 | 4.55 | |
| Supplementary motor area | R | 6 | 4 | 4 | 64 | 5.19 | 4.55 | |
| Supramarginal gyrus | L | 2 | −56 | −28 | 40 | 5.13 | 4.50 | |
| Superior frontal gyrus | L | 6 | −24 | -2 | 58 | 4.73 | 4.22 | |
| Postcentral gyrus | L | 3 | −56 | −20 | 34 | 4.53 | 4.07 | |
| Middle frontal gyrus | L | 6 | −28 | -8 | 52 | 4.48 | 4.04 | |
| Middle temporal gyrus | R | 37 | 48 | −60 | 8 | 5.44 | 4.72 | 569 |
| Cerebelum VIIb | L | | −16 | −74 | −48 | 4.95 | 4.38 | 427 |
| Cerebelum VIII | L | | −24 | −66 | −52 | 4.75 | 4.24 | |
| Negative | | | | | | | | |
| Cuneus | L | 17 | −10 | −96 | 16 | 7.30 | 5.85 | 5904 |
| Cuneus | R | 18 | 14 | −90 | 28 | 6.80 | 5.40 | |
| Lingual gyrus | R | 18 | 14 | −62 | -2 | 6.05 | 5.11 | |
| Lingual gyrus | L | 18 | −18 | −74 | -8 | 5.26 | 4.60 | |
| Calcarine | L | 18 | 0 | −76 | 18 | 5.11 | 4.49 | |
| Fusiform gyrus | L | 18 | −24 | −80 | −16 | 4.95 | 4.38 | |
| Calcarine | R | 17 | 20 | −54 | 6 | 4.72 | 4.22 | |
| Cerebelum Crus I | L | | −38 | −78 | −18 | 4.37 | 3.95 | |
| Middle occipital gyrus | L | 18 | −16 | −86 | -4 | 4.22 | 3.84 | |

DOI: https://doi.org/10.7554/eLife.42265.022

right AIC is related to accuracy in sensing the timing of one's bodily signals, for example, heartbeat (*Critchley et al., 2004*). Consistent with the notion that the AIC contributes to accurate perception of bodily states (*Bechara and Naqvi, 2004*), the insula works as a hub to convey bodily information into internal feelings for maintaining homeostasis and to mediate the representations of visceral states that link to the representations of the external world (*Farb et al., 2013a*).

Our finding of a critical role of the AIC in interoceptive attention fits with a recent predictive coding account of the brain (*Bastos et al., 2012*; *Friston and Kiebel, 2009*; *Rao and Ballard, 1999*),

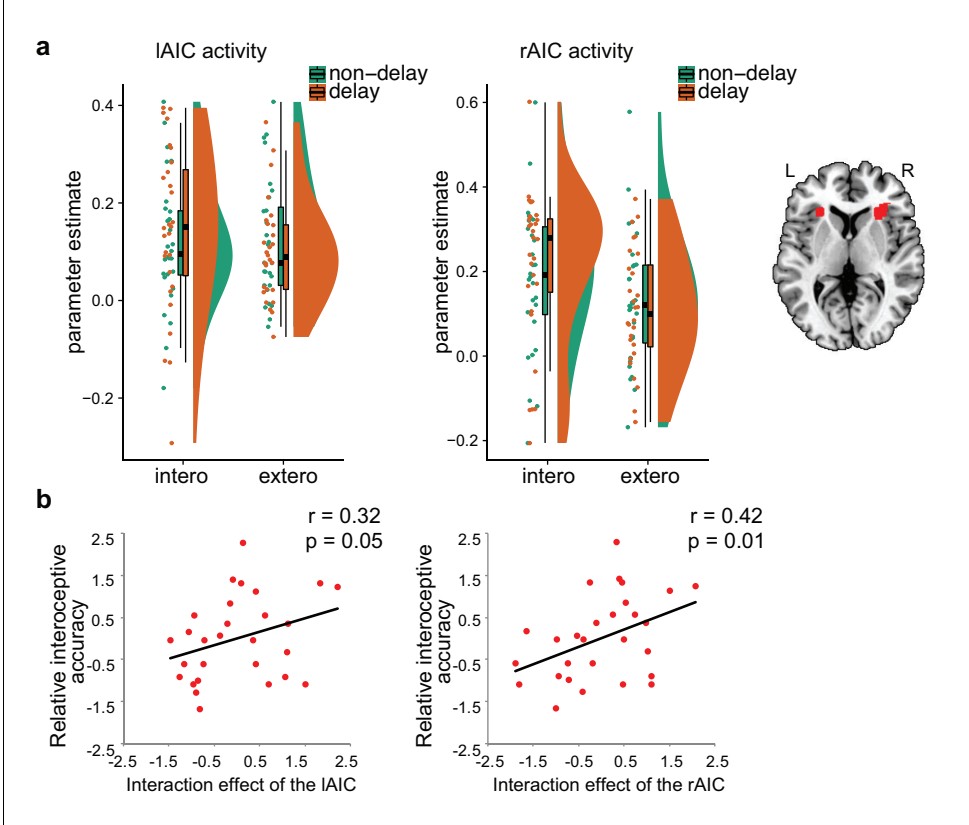

**Figure 5.** ROI results of the second fMRI sample. (**a**) ROI analysis of the parameter estimates of the left and the right AIC under the four experimental conditions. Raincloud plots were used for visualization. (**b**) Correlation between the interaction effect of bilateral AIC and relative interoceptive accuracy. The values of the variable in b were normalized as z-scores.

DOI: https://doi.org/10.7554/eLife.42265.023

The following source data and figure supplements are available for figure 5:

**Source data 1.** CSV file containing data for *Figure 5b*.

DOI: https://doi.org/10.7554/eLife.42265.027

**Figure supplement 1.** Raincloud plot visualization of respiratory volumes under the four experimental conditions from the second fMRI sample.

DOI: https://doi.org/10.7554/eLife.42265.024

**Figure supplement 1—source data 1.** CSV file containing data for *Figure 5—figure supplement 1*.

DOI: https://doi.org/10.7554/eLife.42265.029

**Figure supplement 2.** Activation maps without and with RETROICOR +RVHRCOR correction for the second fMRI sample.

DOI: https://doi.org/10.7554/eLife.42265.025

**Figure supplement 3.** Paired t-test of beta maps obtained without and with RETROICOR + RVHRCOR correction for the second fMRI sample.

DOI: https://doi.org/10.7554/eLife.42265.026

which suggests that the brain actively tries to predict possible future states and to minimize the difference between actual and predicted states. In the context of interoceptive and embodied predictive coding (*Allen and Friston, 2018*; *Gu et al., 2013*), previous studies hypothesized that interoceptive predictions are computed within a network of brain regions with the AIC as the key node (*Allen et al., 2016*; *Barrett and Simmons, 2015*; *Seth, 2013*). Empirical evidence that directly supports this computational role of the insula is still rare. One such study using a tactile oddball paradigm and DCM of fMRI time series demonstrated that the AIC is the only region, among a network of body-related brain regions, that shows a reciprocal increase in connectivity with the somatosensory cortex (*Allen et al., 2016*). Our finding is consistent with this previous study and extends the

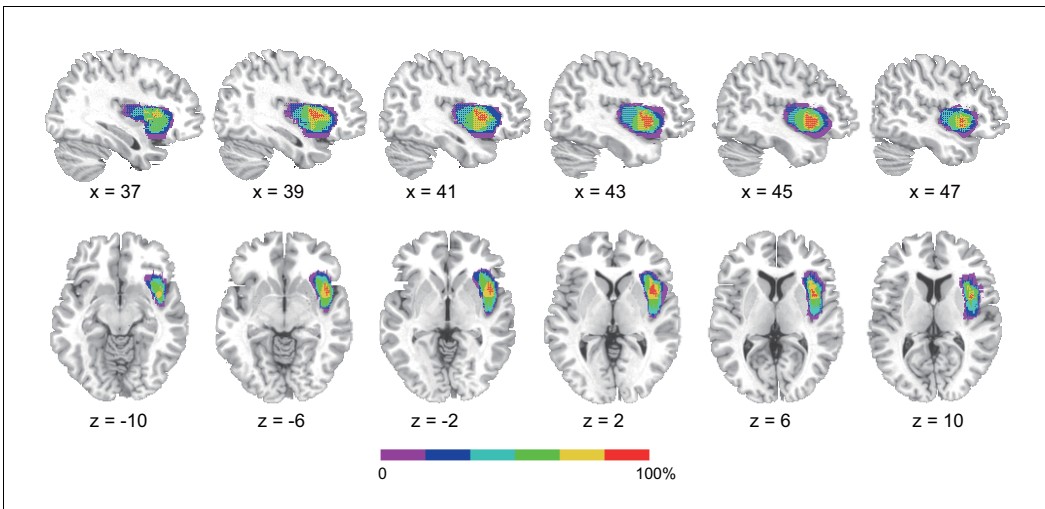

**Figure 6.** Reconstruction of anterior insular cortex lesions of six patients. Red color indicates 100% overlap. Left lesions were flipped to the right side to map the lesion overlap.
DOI: https://doi.org/10.7554/eLife.42265.028

role of the AIC in predictive coding to breathing-related interoception by using both fMRI and lesion approaches.

The role of the AIC in interoceptive attention identified by the fMRI studies was augmented by the data from patients with focal damage to the insula. Relative to non-insular lesion patients and healthy controls, AIC lesions led to a deficit in accuracy and sensitivity of interoceptive attention. These findings provide causal evidence demonstrating the critical role of the AIC in interoceptive attention. Traditionally, the insular cortex is considered as a limbic sensory region that participates in the intuitive processing of complex situations (*Augustine, 1996*; *Butti and Hof, 2010*; *Menon and Uddin, 2010*) by integrating the ascending visceromotor and somatosensory inputs with attention systems via intrinsic connectivity to identify and respond to salient stimuli (*Menon and Uddin, 2010*; *Uddin, 2015*). The AIC, in particular, is a node that mediates cognitive processes including bottom-up control of attention (*Corbetta et al., 2002*; *Corbetta et al., 2008*; *Wu et al., 2015*) and conscious detection of signals arising from the autonomic nervous system (*Craig, 2002*; *Critchley, 2004*). Therefore, the behavioral deficit of interoceptive attention in patients with AIC lesions is due to the disruption in the integration of the somatic and visceral inputs with the abstract representation of the present internal state (i.e., the saliency of a certain type of signals). Consequently, it leads to failure in discriminating whether the displayed respiratory curve is different from internal states.

Most previous lesion studies indicated interoceptive deficits with AIC lesions (*Critchley and Garfinkel, 2017*; *García-Cordero et al., 2016*; *Ibañez et al., 2010*; *Ronchi et al., 2015*; *Starr et al., 2009*; *Terasawa et al., 2015*; *Wang et al., 2014*), supporting the conclusion that interoceptive accuracy relies on a widely distributed network with the insular cortex as a key node (*Craig, 2002*; *Critchley and Harrison, 2013*). However, the preservations of interoceptive processing (*Khalsa et al., 2009*) and self-awareness across a large battery of tests (*Philippi et al., 2012*) were documented in one patient with bilateral insular damages. These studies are mostly based on subjective report focusing on 'feeling/awareness' (*Khalsa et al., 2009*) that might be compensated by other brain structures such as the brainstem and subcortical structures, for example, nucleus tractus solitaries, the parabrachial nucleus, area postrema and hypothalamus (*Damasio et al., 2013*), frontal and temporal regions, for example, amygdala, superior temporal gyrus, and temporal pole (*García-Cordero et al., 2016*; *Shany-Ur et al., 2014*). In the current study, the BDT challenged interoceptive attention that requires the integration of interoceptive awareness and accuracy. Our examination of interoceptive attention in patients with focal AIC lesions showed that lesions of the AIC were associated with a deficit in performance, indicating that the AIC is critical in supporting the precision of interoceptive processing.

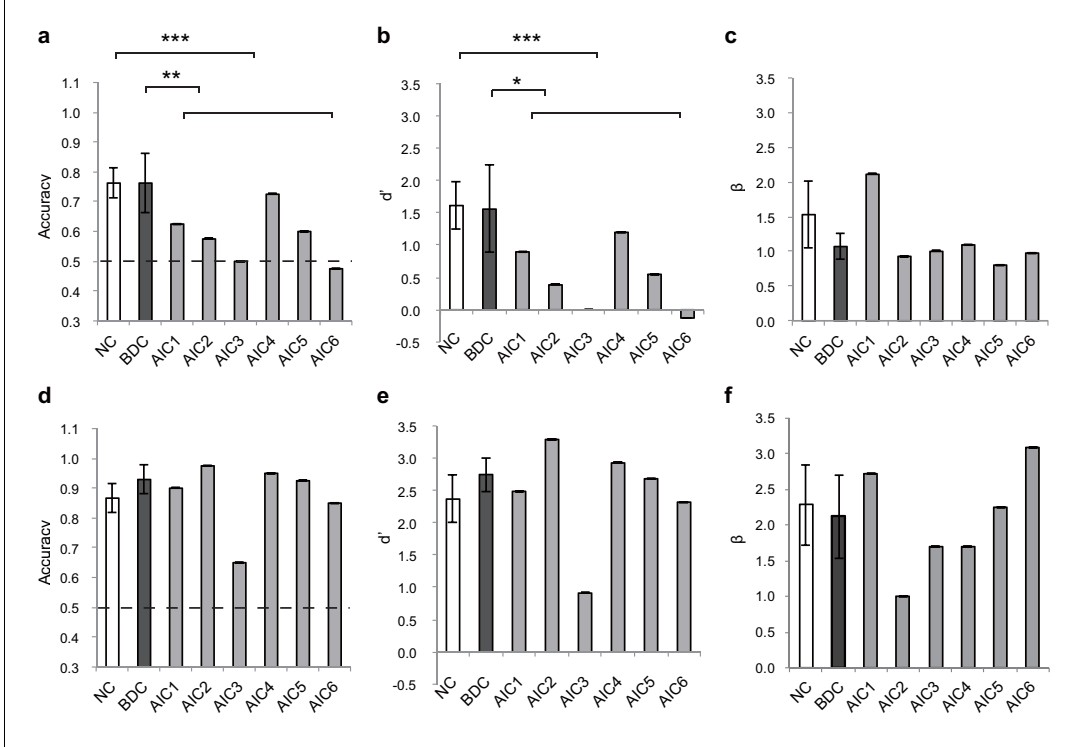

**Figure 7.** Behavioral results of the lesion study. (a, b, c) the interoceptive performance on the BDT, and (d, e, f) the exteroceptive performance on the DDT. On the BDT, patients with AIC lesions had significantly lower performance in accuracy and $d'$ compared with the NC and BDC groups but did not show significant alteration in $\beta$ during the BDT. On the DDT, patients with AIC lesions did not show significant abnormality in performance in accuracy, $d'$, and $\beta$ compared with either the NC or BDC groups. NC, normal control; BDC, brain damage control. Dashed line: chance level. * $p < 0.05$; ** $p < 0.01$; *** $p < 0.001$.

DOI: https://doi.org/10.7554/eLife.42265.030

The following source data is available for figure 7:

**Source data 1.** CSV file containing behavioral data for lesion study.
DOI: https://doi.org/10.7554/eLife.42265.031

## Mechanisms of the AIC in relation to interoceptive attention

Interoceptive attention is the mechanism that coordinates the processing of bodily signals and higher-level representation of that information. The AIC reportedly encodes and represents bodily information (e.g., visceral states) and transmits this information to other neural systems for advanced computations in conscious perception and decision-making (*Bechara and Naqvi, 2004*; *Flynn, 1999*;

**Table 8.** Statistics of the results of the lesion study.

|  |  | Accuracy | | $d'$ | |
|---|---|---|---|---|---|
|  |  | T | BF | T | BF |
| BDT | AIC vs. NC | −3.47*** | 14.71 | −3.62*** | 13.78 |
|  | AIC vs. BDC | −2.35** | 3.95 | −2.22* | 3.40 |
|  | BDC vs. NC | 0 | 0.42 | 0.11 | 0.43 |
| DDT | AIC vs. NC | 0.18 | 0.38 | 0.18 | 0.38 |
|  | AIC vs. BDC | −0.99 | 0.98 | −0.83 | 0.85 |
|  | BDC vs. NC | 1.74* | 0.82 | 1.46 | 0.69 |

* p<0.05; ** p<0.01; *** p<0.001; one- tailed; BF, Bayes factor.
DOI: https://doi.org/10.7554/eLife.42265.032

*Gu and FitzGerald, 2014*). The AIC is a key node of the large-scale network that detects information from multiple sources including objective visceral signals, generates subjective awareness (*Craig, 2009*; *Gu et al., 2012*; *Gu et al., 2015*; *Kleckner et al., 2017*; *Seeley et al., 2007*), and responds to the switch between networks that supports internal oriented processing and cognitive control (*Menon, 2011*; *Menon and Uddin, 2010*; *Sridharan et al., 2008*). Supporting this argument, we showed that the AIC is intrinsically connected to the somatosensory area of the PoCG and that this connection is positively modulated by interoceptive attention (relative to exteroceptive attention).

Other higher-level areas, for example, the ACC/SMA, FEF, and IFG of the so-called cognitive control network (CCN) (*Fan, 2014*; *Wu et al., 2018*), are also involved in the interoceptive process. This is supported by the results of the enhanced functional connectivity between the AIC and these regions. Both somatosensory afferents and a network that includes the AIC and the ACC are possible pathways of interoceptive attention (*Khalsa et al., 2009*). The AIC may play a central role in integrating sensory signals from the PoCG and visual cortex and sends top-down signals that guide sensation and perception through a dynamic interaction with sensory or bottom-up information. Somatosensory information concerning the internal state of the body is conveyed through the PoCG, as well as the visual signals in V2/3 containing the majority of external information. The top-down modulation of the AIC in interoceptive attention is accomplished by augmenting the efferent signals to the somatosensory cortices. This result is consistent with the argument that a first-order mapping of internal feeling is supported by insular and somatosensory cortices (*Damasio, 2003*) and that somatosensory information critically contributes to interoceptive attention (*Khalsa et al., 2009*).

In the BDT, interoceptive attention reflects a combination of the attention to the internal bodily signal (i.e., the breath) and the external visual stimulus (i.e., the curve). To coordinate perceptual processing, the AIC may distribute and balance the processes of external and internal information. The winning model and parameter inference from DCM provide evidence that interoceptive attention is achieved mainly by modulating the connectivity between the AIC and the somatosensory areas (PoCG), while exteroceptive attention is primarily modulated via the connectivity between the AIC and V2/3. We propose that the dynamic adjustment of the connectivity of the AIC to sensory cortices is the foundation of interoceptive attention for bodily signals, which is critical for homeostatic regulation, and of exteroceptive attention for external objects or inputs.

## Interoceptive task in the respiratory domain

Although the neural correlates of interoceptive awareness have been studied by other tasks, such as the heartbeat detection task (*Bechara and Naqvi, 2004*; *Critchley et al., 2004*; *Khalsa et al., 2009*; *Ring et al., 2015*), the error rate arising from the difficulty in heartbeat counting or non-sensory process confounds are inherent to these cardioception designs (*Kleckner et al., 2015*; *Ring et al., 2015*). In contrast to cardioception, breath can be clearly perceived and autonomously controlled. This feature enabled us to design a task measuring interoceptive attention, which requires that the target of interoception be clearly and vividly perceivable by our consciousness. The positive correlations between objective interoceptive accuracy during the BDT and subjectively scored difficulty of interoceptive task relative to exteroceptive task (i.e., interoceptive awareness) further demonstrates that the BDT is valid in assessing interoceptive attention. We developed the respiratory interoception task as a non-intrusive measurement with low cognitive load that is more practical for patients with focal brain damage than the demanding cardiac interoception tasks.

The BDT may not represent a pure probe of interoception because respiratory processes can also be tracked using exteroceptive and proprioceptive information. Thus, the participants possibly relied on a mix of interoceptive, exteroceptive, and proprioceptive information to perform the task. In our design, we included the DDT for a measure of exteroception so that the cognitive subtraction of DDT from BDT leaves the interoceptive and proprioceptive processing components of interoception (*Gu et al., 2013*). In the BDT, the delayed manipulation in our study was fixed to 400 ms, approximately 1/10 of an average cycle of normal healthy people (i.e., 3–4 s/cycle). This delayed duration can be manipulated according to each individual's respiratory cycle in an effort to control subjective task difficulty across participants.

## Interoceptive attention

Depending on the source of information, attention can be categorized into (1) interoceptive attention, which is directed toward bodily signals such as somatic and visceral signals (e.g., in a heartbeat detection or counting task); (2) exteroceptive attention, which is directed toward primary sensory inputs from outside (e.g., visual and auditory stimuli); and (3) executive control of attention, which coordinates thoughts and actions (e.g., in Color Stroop, flanker, and working memory tasks; see review *Fan, 2014*). Although extensive studies have focused on the attentional modulation of sensory and perceptual inputs and on the executive control of attention, interoceptive attention is difficult to study because the vast majority of intrinsic visceral activity, except breath effort, cannot be clearly perceived under normal conditions. Using the BDT to examine attentional deployment toward breath effort enabled us to reveal the neural mechanism of interoceptive attention. In general, the perceptible, controllable, measurable, and autonomous features of breathing guaranteed more accurate and reliable measurement of individual differences in interoceptive ability.

As a type of interoceptive attention that could be clearly perceived and autonomously controlled, breath plays a potentially important role in generating and regulating emotion. For example, mindfulness meditation, which is now well known for its role in emotion regulation and mental health (*Khoury et al., 2015*), can be viewed as a practice involving interoceptive attention. One of its primary methods is to bring one's attention (the processing) and then awareness (the outcome) to the current experiences of the movement of the abdomen when breathing in and out or the breath as it enters and exits the nostrils. Considering the revealed neural mechanisms of interoceptive attention in this study, we predict that the AIC plays an important role in meditation. Findings that meditation experience is associated with increased gray matter thickness in the AIC (*Lazar et al., 2005*) and increased gyrification (increase in folding) of the AIC (*Luders et al., 2012*) support this prediction. Meditation training may enhance interoceptive attention to focus on bodily signals so that accurate feelings can be generated based on the bodily responses, meanwhile the mind can be released from an intensive involvement of exteroceptive and executive control of attention (the internal attention for the coordination of thought processing) that consumes the majority of mental resources.

## Conclusion

This study provided important evidence of the involvement of the AIC in interoceptive attention by the fMRI studies and further demonstrated that the AIC is critical for the process by the lesion study. The converging evidence also suggests that interoceptive attention is achieved through top-down modulation from the AIC to the somatosensory and sensory cortices. In addition, the implementation of the interoceptive task extends the research on interoceptive processing into the respiratory domain with the validity and reliability demonstrated. It may have significant applications in studying issues related to interoceptive attention in patients with neuropsychiatric disorders, such as anxiety (*Avery et al., 2014*) and autism (*Barrett and Simmons, 2015*; *Quattrocki and Friston, 2014*), and in patients with substance use disorders (*Sönmez et al., 2017*).

# Materials and methods

## Task design

### Task implementations

A respiratory transducer (TSD201, MRI compatible, BIOPAC Systems Inc), which was fastened around the participants' upper chest, was utilized to record breathing effort by measuring thoracic changes in circumference during respiration. The signal for the change in circumference was sampled at 1000 Hz using the BIOPAC MP150/RSP100C system, passing through a DC amplifier with low-pass filtering at 1 Hz and high-pass filtering at 0.05 Hz, and gain set to 10 V. Analog signal was then digitized by an A/D converter (USB-1208HS-4AO, Measurement Computing, Inc) and sent to a USB port of the test computer (*Figure 1a*). The task program in E-Prime (Psychology Software Tools, Pittsburgh, PA, USA) served as an interface through which the digitized signal from the USB port was received and presented to the participants on the computer screen as a continuous blue breath curve extending from left to right as time elapsed (*Figure 1b*), which was representative of their breathing effort. The breath curve was presented either with or without a delay (*Figure 1c*).

For the engagement of interoceptive attention during BDT, the participants were required to judge whether the presented breath curve was delayed relative to the breath rhythm they perceived from their body. In half of the trials, the displayed breath curve was synchronized with the participant's own respiration. In the other half, the displayed breath curve was delivered after a 400 ms delay period relative to the participant's own respiration (i.e., the plotting of the point on the extending curve was actually the point saved 400 ms before the current time point). Note that the parameter of 400 ms delay was determined based on a proportion (~1/10) of an average respiratory cycle of normal healthy people (i.e., 3–4 s/cycle). For the engagement of exteroceptive attention, the DDT was performed. The participants were instructed to detect whether a red dot flashed on the respiratory curve at any time when the breath curve was displayed. In half of the trials, a red dot flashed (30 ms for the fMRI experiment, and 50 ms for the lesion study) at a randomized time point on the breath curve. *Figure 1b* illustrates the two tasks. The stimuli for these two tasks were the same, consisting of four trial types reflecting the combination of presence or absence of a delay and presence or absence of a dot (*Figure 1c*). The factor of attentional deployment involved directing attention interoceptively to 'respiration' or exteroceptively to 'dot' in the BDT and DDT, respectively. During the two tasks, the participants were instructed to breathe as usual without holding or forcing their breath.

The participants were asked to perform each task in a blocked fashion in the interoceptive and exteroceptive runs. The fMRI experiment consisted of two runs, with one run for the BDT and the other run for the DDT. Each run, which included 60 trials, began and ended with a 30 s blank display, and each trial lasted 18 s, with an average inter-trial interval of 2 s, for a total of 21 min per run. Each trial began with a 3 s "Relax" display, followed by a 12 s respiratory curve presented with or without a 400 ms delay, and ended with a 3 s response window during which the participants made a forced-choice button-press response, prompted by the presentation of two alternative choices to indicate their response (*Figure 1b*). After the fMRI scan, the participants were asked to indicate the subjective difficulty they felt for each task on a 1–10 scale, with higher value indicating higher difficulty. For the lesion study, the same tasks were employed, one run for each task, with 40 trials in each run.

## Behavioral data analysis

Interoceptive attention is associated with the objective *accuracy* in detecting bodily signals, the subjective belief in one's ability to detect bodily signals in general (i.e., *sensibility*), and the correspondence between objective accuracy and subject report (i.e., metacognitive *awareness* about one's performance when detecting bodily signals) (*Garfinkel and Critchley, 2013*; *Garfinkel et al., 2015*). Objective interoceptive/exteroceptive accuracy was calculated as the overall correct response rate during the BDT/DDT. In addition, we used signal detection theory to index detection sensitivity and response bias. Signal detection theory characterizes how perceivers separate signal from noise, assuming that the perceiver has a distribution of internal responses for both signal and noise (*Snodgrass and Corwin, 1988*; *Stanislaw and Todorov, 1999*). A fundamental advantage of signal detection theory is the distinction between sensitivity (ability to discriminate alternatives) and bias (propensity to categorize 'signal' or 'noise'). For the BDT, the sensitivity index ($d'$) was calculated as $d'=Z_{hit\ rate} - Z_{false\ alarm\ rate}$, where the hit rate is the proportion of trials with delayed breath curve and responded as 'yes', and the false alarm rate is the proportion of trials with non-delayed breath curve and responded as 'yes'. A higher value of $d'$ indicates a better interoceptive accuracy, whereas a value of 0 represents that the performance is at the chance level. The response bias index $\beta$, which represents the position of the subjective decision criterion, was defined as $\beta = \exp(d' \times C)$, where $C = -(Z_{hit\ rate} + Z_{false\ alarm\ rate})/2$. Index $\beta$ corresponds to the distance of participants' estimated criterion to ideal observer criterion, and a value of 1 indicates no bias. For the DDT, indices of $d'$ and $\beta$ were calculated using the same formula, with the dot present as 'signal' and dot absent as 'noise'. Relative interoceptive accuracy was defined as the difference in performance accuracy between the BDT and DDT to control for non-specific performance effects (*Critchley et al., 2004*).

An individual's subjective account of how they experience internal sensation and perception represents an alternative aspect of interoceptive processing, namely sensibility (*Garfinkel et al., 2015*). In the fMRI experiment, the subjective sensibility of interoceptive processing was measured using the self-report questionnaire of Body Perception Questionnaire (BPQ) (*Porges, 1993*). The subjective perception of one's performance during the BDT represents the awareness aspect of

interoceptive attention (*Garfinkel et al., 2015*), which was measured via the subjectively scored difficulty of the BDT relative to the DDT. Note that the confidence ratings of the performance would be more directly related to awareness of interoception, but the measures of subjective difficulty and confidence ratings should be closely related. The correlation between the relative interoceptive accuracy and these indices of subjective sensibility and awareness was calculated to examine the relationship between the perceived (subjective) and actual measured (objective) performance of interoceptive attention. In addition, we conducted BFs of these correlation coefficients using JASP (*The JASP Team, 2018*). A BF larger than three suggests a significant correlation, whereas a BF smaller than 1/3 indicates a null correlation.

## fMRI experiments

### Participants

The fMRI experiments included two samples of participants: the first sample included 44 adults (23 females and 21 males, mean age ± standard deviation: 21.43 ± 2.51 years, age range: 19–29 years), and the second sample included additional 28 adults (14 females and 14 males, mean age ± standard deviation: 21.93 ± 2.11 years, age range: 18–26 years). All participants underwent the same experimental procedures, except that pulse and respiratory signals were recorded for the second sample using the pulse sensor (Siemens Peripheral Pulse Unit, PPU_098) of the scanner and BIOPAC, respectively. All participants were right-handed (except for one participant), reported normal or corrected-to-normal vision, and had no known neurological or visual disorders. All participants completed questionnaires indexing subjective interoceptive sensibility (BPQ), symptoms of anxiety (Hamilton anxiety scale, HAMA; *Hamilton et al., 1976*), depression (Beck Depression Inventory, BDI; *Knight, 1984*), and positive and negative affective experience (PANAS) (*Watson, 1988*). They provided written informed consent in accordance with the procedures and protocols approved by The Human Subjects Review Committee of Peking University.

### fMRI data acquisition and preprocessing

During functional scanning, the participants performed the BDT and DDT in separate runs that required them to attend to either their respiration or a visual flash dot, respectively. All neuroimaging data were acquired on a MAGNETOM Prisma 3T MR scanner (Siemens, Erlangen, Germany) with a 64-channel phase-array head-neck coil. During the tasks, blood oxygen level-dependent (BOLD) signals were acquired with a prototype simultaneous multi-slices echo-planar imaging (EPI) sequence (echo time, 30 ms; repetition time, 2000 ms; field of view, 224 mm ×224 mm; matrix, 112 × 112; in-plane resolution, 2 mm ×2 mm; flip angle, 90 degree; slice thickness, 2.1 mm; gap, 10%; number of slices, 64; slice orientation, transversal; bandwidth, 2126 Hz/Pixel; slice acceleration factor, 2). For the second cohort, the thickness was changed to 2 mm with a gap of 15%, and the number of slices was changed to 62. Field map images were acquired using a vendor-provided Siemens gradient echo sequence (gre field mapping: echo time 1, 4.92 ms; echo time 2, 7.38 ms; repetition time, 635 ms; flip angle, 60 degree; bandwidth, 565 Hz/Pixel) with the same geometry and orientation as the EPI image. A high-resolution 3D $T_1$ structural image (3D magnetization-prepared rapid acquisition gradient echo; 0.5 mm ×0.5 mm × 1 mm resolution) was also acquired. Image preprocessing was performed using Statistical Parametric Mapping package (SPM12, RRID: SCR_007037; Welcome Department of Imaging Neuroscience, London, United Kingdom). EPI volumes were realigned to the first volume, corrected for geometric distortions using the field map, coregistered to the $T_1$ image, normalized to a standard template (Montreal Neurological Institute, MNI), resampled to 2 × 2 × 2 mm$^3$ voxel size, and spatially smoothed with an isotropic 8 mm full-width at half-maximum Gaussian kernel.

### fMRI: analysis of the first sample

#### Image statistical parametric mapping

Imaging data from the two samples were analyzed separately and independently, with the exploratory whole brain analysis conducted with the first sample and the confirmatory ROI analysis conducted with the second sample. For the whole brain analysis of the first sample, statistical inference was based on a random-effects approach (*Penny and Holmes, 2007*), which comprised two steps: first-level analyses estimating contrasts of interest for each subject followed by second-level analyses

for statistical inference at the group level. For each participant, first-level statistical parametric maps of BOLD signals were modeled using general linear modeling (GLM) with regressors defined for each run with the four trial types: 2 breath curve delay (non-delayed, delayed)×2 dot present (no dot, dot). Each trial was modeled as an epoch-related function by specifying an onset time and a duration of 12 s. The corresponding four regressors were generated by convolving the onset of each trial with the standard canonical hemodynamic response functions (HRF) with a duration of 12 s, that is, by convolving the trial block with HRF, equivalent to a box-car function. Six parameters generated during motion correction were entered as covariates of no interest. The time series for each voxel were high-pass filtered (1/128 Hz cutoff) to remove low-frequency noise and signal drift.

Contrast maps for interoceptive vs. exteroceptive attention (BDT – DDT), the presence of breath curve delay (delayed – non-delayed), and the interaction between them ([delayed – non-delayed] $_{BDT}$ – [delayed – non-delayed] $_{DDT}$) for each participant were entered into a second-level group analysis conducted with a random-effects model that accounts for inter-subject variability and permits population-based inferences. The statistical maps were corrected for multiple comparisons using Gaussian random field (GRF) theory (T > 3.29, cluster-wise p<0.05, GRF corrected) with a minimum cluster size of 420 resampled voxels. Note that changes in neural activity revealed by the main effect of interoceptive vs. exteroceptive attention (the contrast of BDT vs. DDT) could also reflect task-specific effects, such as differences in task difficulty or respiratory characteristics (i.e., amplitude and frequency) between the two tasks, in addition to effects of change in attentional focus. Although the main effect of interoceptive vs. exteroceptive attention (the contrast of BDT vs. DDT) is subject to confounding by the task-specific effects, the interaction effect can disentangle those effects (i.e., cancel out the breathing effort difference between the two tasks). This interaction reflected the brain response when directing attention to the feedback mismatch during interoceptive processing while controlling for the non-specific effect (i.e., the physical difference in feedback stimulus between delayed and non-delayed curves during exteroceptive processing). Therefore, a positive interaction effect represents brain response to the interoceptive processing above and beyond the physical feedback difference.

## Correlation between interoceptive accuracy and the interaction effect of the AIC

To test for a linear correlation between AIC activation and behavioral performance on the BDT, we entered each participant's interaction contrast maps into the second-level random-effects group regression analysis, together with their individual accuracy in the BDT as the variable of interest and accuracy in the DDT as the covariate. Threshold of significance was GRF-corrected at p<0.05 (T > 2.42) with a cluster extent of 106 contiguous voxels (resampled), corrected using small-volume ROI correction. The mask image was generated from an anatomical template of the bilateral insular cortex based on the Automated Anatomical Labeling template (*Tzourio-Mazoyer et al., 2002*).

## PPI analysis

PPI analysis provides a measure of change in functional connectivity between different brain regions under a specific psychological context (*Friston et al., 1997*). We conducted PPI analyses using a moderator derived from the product of the activity of a seed region (i.e., the AIC) and the psychological context (i.e., interoceptive in contrast to exteroceptive attention, BDT vs. DDT). The ROI selection was independent of the interoceptive attention process that was used as the psychological context: The left and right AICs were first identified from the main effect of the breath curve delay (the contrast of delayed versus non-delayed) in the GLM. We then conducted two whole-brain PPI tests for the right and left AIC, reflecting changes in functional connectivity between the seed region time series (physiological regressor) and other brain regions as a function of interoceptive relative to exteroceptive attention (BDT vs. DDT, psychological regressor). The AIC time series of each participant were extracted from a 6 mm-radius sphere centered at the peak of the AIC (right AIC: x = 30, y = 26, z = −4; left AIC: x = −30, y = 24, z = −4). The PPI term was calculated as an element-by-element product of the deconvolved physiological regressor and psychological regressor, which was then reconvolved with the canonical HRF. The generated PPI model included the PPI term, the physiological regressor, the psychological regressor, and nuisance regressors of six motion parameters. The threshold of significance for the second-level group data analysis of the images from the PPI

regressor was determined the same as in the GLM. Regions identified as significant clusters have two possible interpretations: (1) the connectivity between the AIC and these regions was altered by the psychological context, or (2) the response of these regions to the psychological context was modulated by AIC activity. To simplify the explanation, we used the first interpretation throughout this article.

## DCM analysis

DCM (*Friston et al., 2003*) is used to disambiguate different potential network structures by inferring hidden neuronal states from measurements of brain activity. DCM distinguishes between endogenous coupling and context-specific coupling, which could account for the effects of experimentally controlled network perturbations. Considering the inherent limited causal interpretability of the PPI analysis for the direction of interaction, we only conducted DCM to explain the potential mechanisms of interplay between AIC and other brain areas involved in interoceptive attention. The ROI of the right AIC in the DCM was the same as that in the PPI analysis. The other regions included in the DCM were selected based on significant positive and negative PPI results and with the coordinates of the ROIs identified by the group level T-contrast of all conditions versus baseline. Data from one participants were excluded from the DCM analysis because activity in one of the ROIs could not be identified.

A three-area DCM was specified for all participants with bidirectional endogenous connection between the right AIC and the other two ROIs, and with the main effect of 'all stimuli' as the driving input entering the other two ROIs. Five base models were generated by specifying possible modulations of interoceptive and exteroceptive attention (BDT and DDT, respectively) on the four endogenous connections between ROIs. These base models were then systematically elaborated to produce 52 variant models, which included all possible combinations of the modulation of interoceptive and exteroceptive attention (BDT and DDT, respectively) on endogenous connections between the right AIC and the two other ROIs.

Model comparison was implemented using random-effects BMS in DCM12 to determine the most likely model of the 52 models given the observed data from all participants (*Stephan et al., 2009*). The RFX analysis computes exceedance and posterior probabilities at the group level, and the exceedance probability of a given model denotes the probability that this model is more likely than all other models considered (*Stephan et al., 2009*). To summarize the strength of effective connectivity and its modulation quantitatively, we used random-effects BMA to obtain average connectivity estimates (weighted by their posterior model) across all models and all participants (*Penny et al., 2010*). We conducted one-sample t tests on the subject-specific BMA parameter estimates to assess their consistency across subjects with Bonferroni correction for multiple comparisons.

## fMRI: ROI analyses of the second sample

Whereas whole brain analyses of the first sample aimed at identifying brain areas involved in interoceptive processing, ROI analyses of the second sample aimed to confirm that the effects found from the first sample were not confounded with the effects induced by other physiological signals. Change in BOLD signals can be due to direct neural activity (induced by experimental manipulation) or an indirect effect (such as vascular response, which would be considered as a confounding effect). For example, the cerebral vascular response is sensitive to the circulation of carbon dioxide ($CO_2$) and oxygen ($O_2$), and causes a change in global cerebral blood flow (CBF) and global BOLD signal. It is evident both in human and animals that the global CBF and global BOLD responses influence local stimulus-induced hemodynamic response to neural activation (*Cohen et al., 2002*; *Friston et al., 1990*; *Ramsay et al., 1993*). In general, a larger local stimulus-induced BOLD response occurs when global BOLD is lowered, whereas a smaller local stimulus-induced BOLD response occurs when global BOLD is elevated. In our study, the experimental manipulation of interoception was likely to cause a change in respiratory characteristics (i.e., circulation of $CO_2$ and $O_2$). The difference in physiologic states between the BDT and the DDT might cause a change in global BOLD signals. Thus, the effect resulting from local interoception-related BOLD responses would be confounded by the global hemodynamic influence.

To partial out the potential confounding, we processed physiological data, including cardiac pulsation and respiratory volume collected in the second sample, by using the PhLEM toolbox (http://sites.google.com/site/phlemtoolbox/). Physiological noise correction consisted of (1) regressing out time-locked cardiac and respiratory effects, and their interaction effect using a modification of the conventional RETROICOR approach (*Brooks et al., 2008*; *Glover et al., 2000*), and (2) regressing out low-frequency respiratory and heart rate effects using the RVHRCOR approach (*Verstynen and Deshpande, 2011*). In RETROICOR, a cardiac phase calculated from a pulse oximeter was assigned to each acquired image in a time series (*Hu et al., 1995*), and a respiratory phase was assigned to a corresponding image using the histogram equalized transfer function that considers both the respiratory timing and depth of breathing (*Glover et al., 2000*). The conventional RETROICOR approach (*Glover et al., 2000*) defines low-order Fourier terms (i.e., sine and cosine values of the principal frequency and the $2^{nd}$ harmonic) to model the independent effects of the cardiac and respiratory fluctuation, which is considered insufficient to remove variations caused by physiological artifacts (*Harley and Bielajew, 1992*; *Tijssen et al., 2014*). Therefore, we used additional terms of higher-order Fourier expansions (i.e., to the 5th harmonics) in RETROICOR, and formed multiplicative sine/cosine terms that consider the interaction between cardiac and respiratory effects. In specific, the interaction terms were calculated by $\mathrm{Sin}(\varphi_c \pm \varphi_r)$ and $\mathrm{Cos}(\varphi_c \pm \varphi_r)$, where $\varphi_{c,r}$ is the cardiac or respiratory phase, consisting of a mixture of third-order cardiac and second-order respiratory harmonics. In RVHRCOR, two nuisance regressors were generated by convolving respiratory variations (RVs) and heart rate (HR) with 'respiration response function (RRF)' and the 'cardiac response function (CRF)' respectively. In specific, RV was computed as the root-mean-square amplitude of the respiration waveform across a 6 s sliding window, and HR was computed as the inverse of the average beat-to-beat interval in a 6 s sliding window (*Chang et al., 2009*). Therefore, the physiological correction contained a total of 46 regressors, of which 20 were from independent time-locked cardiac and respiratory effects, 24 were from interaction terms, and two were from low-frequency RV and HR effects. Statistical parametric maps were generated using the same GLM as in the whole brain analyses, with motion parameters and these physiological regressors entered as covariates of no interest.

To avoid double dipping, we defined the ROIs based on the first sample. In specific, the ROIs (i.e., left and right AICs) were the clusters of the second-level group analysis results of the interaction effect ([delayed – non-delayed] BDT – [delayed – non-delayed] DDT). Parameter estimates were extracted from each ROI in the second sample under the four experimental conditions of each participant, and then entered into a two-way repeated-measures ANOVA. We also examined correlations between the interaction effect of each ROI and behavior measures (i.e., relative interoceptive accuracy) across participants.

To further examine the degree to which physiological correction impacted the whole brain activation, we conducted whole brain paired t-tests by comparing the contrast maps without and with physiological correction at an extremely permissive threshold (voxelwise p<0.05 uncorrected).

## Lesion study

### Brain lesion patient and control groups

Six male patients (33–53 years old, mean 42.17 ± SD 7.31 years) with focal unilateral insular cortex lesions participated in the lesion study (see *Table 9* for patient characteristics). Two patients had a right-side lesion, and four patients had a left-side lesion. In addition, six patients with focal lesions in regions other than the insular cortex (i.e., temporal pole, n = 3, lateral frontal cortex, n = 2, and superior temporal gyrus, n = 1) were recruited as BDCs, and 12 neurologically intact participants were recruited as NCs. All lesions were resulted from the surgical removal of low-grade gliomas. All patients were recruited from the Patient's Registry of Tiantan Hospital, Beijing, China. NC participants were recruited in the local community. All NC participants were right-handed, had normal color vision, and reported no previous or current neurological or psychiatric disorders. BDC patients matched with patients with insular cortex lesions in chronicity ($t(10) = -0.36$, p=0.38, BF = 0.48), and neither group significantly differed from the NC group in age (AIC vs. NC: $t(9) = -1.01$, p=0.18, BF = 0.61; BDC vs. NC: $t(10) = -1.80$, p=0.06, BF = 1.25) nor education (AIC vs. NC: $t(10) = 0.77$, p=0.25, BF = 0.52; BDC vs. NC: $t(6) = -1.04$, p=0.18, BF = 0.72). All six insular lesion patients were considered cognitively intact, as determined by Mini-Mental State Examination (MMSE), a

measurement of cognitive impairment (*Folstein et al., 1975*), and the raw scores of MMSE were not significantly different from either BDCs ($t(10) = -1.30$, p=0.13, BF = 0.78) or NCs ($t(10) = -1.76$, p=0.06, BF = 1.17). Compared with the NCs, the patients with insular lesions did not show significant alteration in baseline mood indexed by the BDI score, compared to NCs ($t(6) = 1.70$, p=0.06, BF = 1.84) or BDCs ($t(9) = 0.65$, p=0.28, BF = 0.53). Demographic information of the groups can be found in *Table 9*. By chance, all the patients with AIC lesions were male. All participants were informed of the study requirements and provided written consent prior to participation. The patient study was approved by the Institutional Review Board of the Beijing Tiantan Hospital, Capital Medical University.

## Lesion reconstruction

Two neurosurgeons, blinded to the experimental design and behavioral results, identified and mapped the lesions of each patient onto a template derived from a digital MRI volume of a normal control (ch2bet.nii) embedded in the MRIcro program (RRID: SCR_008264; http://www.cabiatl.com/mricro/mricro/index.html). In each case, lesions evident on MRI were transcribed onto corresponding sections of the template to create a volume of interest image. This volume of interest image was then used to measure the location (in MNI coordinates) and volume (in mL) of individual lesions and to create within-group overlaps of lesions using the MRIcro program.

## Behavioral data analysis of the lesion study

We used non-parametric analysis (*Feys, 2016*) to test the two-way interaction between group (AIC, BDC, and NC) and task (BDT and DDT) using R (*R Development Core Team, 2013*) because the small sample data sets did not meet the assumption of parametric tests. In specific, we used the *npIntFactRep* function (from the npIntFactRep package) that yielded an aligned rank test for interaction in the two-way mixed design with the group (AIC, BDC, and NC) as the between-subject factor and with the task (BDT and DDT) as the within-subject factor. If the interaction was significant, we used the non-parametric bootstrapping method to test the simple between group effects for each task separately. The bootstrapping procedure was conducted with 10,000 iterations (*Hasson et al., 2003*; *Mooney and Duval, 1993*). If the probability of obtaining the observed t-value was less than 5% (one-tailed), we considered the difference between the two groups to be significant. We used one-tailed tests because we hypothesized that lesions of a specific brain region (i.e., the AIC) would induce deficits in behavioral response. In addition, we calculated BFs with Cauchy prior to determine the relative strength of evidence for the null and alternative hypotheses (*Dienes, 2014*; *Dienes and Mclatchie, 2018*). The value of BF means that the data are BF times more likely under the alternative than under the null hypothesis. The standard value for assessing substantial evidence for the null is BF <1/3 and for the theory against null is BF >3, whereas values between 1/3 and 3 are counted as data insensitivity. The BFs were calculated using JASP (*The JASP Team, 2018*).

**Table 9.** Demographic characteristics of the participants in lesion experiment.

|  | Lesion laterality | Lesion size (ml) | Chronicity (months) | Age (years) | Gender | Education (years) | MMSE | BDI |
|---|---|---|---|---|---|---|---|---|
| IC1 | Right | 3.7 | 38 | 39 | M | 15 | 28 | 4 |
| IC2 | Right | 5.5 | 6 | 33 | M | 16 | 28 | 1 |
| IC3 | Left | 11.2 | 9 | 38 | M | 12 | 26 | 4 |
| IC4 | Left | 9.0 | 12 | 53 | M | 12 | 26 | 8 |
| IC5 | Left | 16.0 | 6 | 51 | M | 16 | 29 | 1 |
| IC6 | Left | 9.2 | 37 | 40 | M | 16 | 26 | 0 |
| BDC | 3 Left/3 Right | 18 ± 14 | 21 ± 16 | 39 ± 7 | 3F/3M | 12 ± 3 | 28 ± 1 | 2 ± 2 |
| NC | N/A | N/A | N/A | 46 ± 7 | 8F/4M | 14 ± 2 | 28 ± 1 | 1 ± 1 |

IC, insular cortex; BDC, brain damage control; NC, normal control; MMSE, mini-mental state examination; BDI, Beck depression inventory.
DOI: https://doi.org/10.7554/eLife.42265.007

## Acknowledgements

We thank Dr. Thomas Beck and Dr. Tian-Yi Qian from Siemens Healthcare for providing the simultaneous multi-slice EPI sequence for fMRI data acquisition. This work was supported by the National Natural Science Foundation of China (grant number: 81729001, 81328008) to JF and ZG. JF was also supported by the National Institute of Mental Health of the National Institutes of Health (NIH) under Award Number R01 MH094305. YW was supported by the research grant of 973 (grant number: 973-2015CB351800) and National Natural Science Foundation of China (grant number: 31771205, 61690205). YY and HG were supported by the Intramural Research Program, National Institute on Drug Abuse, NIH. XW and PL were supported by Beijing Municipal Science and Technology Commission (grant number: Z161100002616014). XW was also supported by the National Natural Science Foundation of China (grant number: 81600931), Beijing Municipal Administration of Hospital' Youth Programs (code: QML20170503) and Capital Health Development Research Project of Beijing, China (grant number: 2016-4-1074). QW was supported by China Postdoctoral Science Foundation (grant number: 2016M600835). XG was supported by the National Institute on Drug Abuse (grant number 1R01DA043695) and the Mental Illness Research, Education, and Clinical Center (MIRECC; VISN3), James J Peters VA Medical Center, Bronx, NY. Dr. Nicholas Van Dam and Evelyn Ramirez were involved at the early stage of the study on interoception. We also thank Shira Russell-Giller and Liat Kofler for their help on proof reading. The authors also thank the National Center for Protein Sciences at Peking University in Beijing, China, for assistance with the MRI data acquisition.

## Additional information

### Funding

| Funder | Grant reference number | Author |
|---|---|---|
| Beijing Municipal Administration of Hospitals | Youth Program QML20170503 | Xingchao Wang |
| National Natural Science Foundation of China | 81600931 | Xingchao Wang |
| Capital Health Development Research Project of Beijing | 2016-4-1074 | Xingchao Wang |
| Brain Research Project of Beijing | Z16110002616014 | Xingchao Wang Pinan Liu |
| China Postdoctoral Science Foundation | 2016M600835 | Qiong Wu |
| National Institute on Drug Abuse | 1R01DA043695 | Xiaosi Gu |
| The Mental Illness Research, Education, and Clinical Center, James J. Peter Veterans Affairs Medical Center | MIRECC VISN 2 | Xiaosi Gu |
| National Institute on Drug Abuse | Intramul Research Program | Hong Gu Yihong Yang |
| National Basic Research Program of China (973 Program) | 973-2015CB351800 | Yanhong Wu |
| National Natural Science Foundation of China | 31771205 | Yanhong Wu |
| National Natural Science Foundation of China | 61690205 | Yanhong Wu |
| National Natural Science Foundation of China | 81328008 | Zhixian Gao Jin Fan |
| National Natural Science Foundation of China | 81729001 | Zhixian Gao Jin Fan |
| National Institute of Mental Health | R01MH094305 | Jin Fan |

The funders had no role in study design, data collection and interpretation, or the decision to submit the work for publication.

## Author contributions

Xingchao Wang, Data curation, Formal analysis, Funding acquisition, Investigation, Methodology, Writing—original draft, Writing—review and editing; Qiong Wu, Data curation, Formal analysis, Investigation, Methodology, Writing—original draft, Writing—review and editing; Laura Egan, Hong Gu, Methodology, Writing—original draft,Writing—review and editing; Xiaosi Gu, Yihong Yang, Methodology, Writing—original draft, Writing—review and editing; Pinan Liu, Funding acquisition, Investigation, Writing—review and editing; Jing Luo, Resources, Writing—original draft,Writing—review and editing; Yanhong Wu, Funding acquisition, Supervision, Investigation, Methodology, Writing—original draft, Writing—review and editing; Zhixian Gao, Data curation, Funding acquisition, Supervision, Investigation, Writing—original draft, Writing—review and editing; Jin Fan, Conceptualization, Supervision, Funding acquisition, Investigation, Methodology, Writing—original draft, Writing—review and editing

## Author ORCIDs

Xingchao Wang ⓘ https://orcid.org/0000-0002-1281-1973
Qiong Wu ⓘ http://orcid.org/0000-0003-2597-4322
Xiaosi Gu ⓘ https://orcid.org/0000-0002-9373-987X
Jin Fan ⓘ https://orcid.org/0000-0001-9630-8330

## Ethics

Human subjects: All participants in fMRI study and in lesion study were gave written informed consent in accordance with the procedures and protocols approved by The Human Subjects Review Committee of Peking University and by The Institutional Review Board of the Beijing Tiantan Hospital, Capital Medical University, respectively.

## Decision letter and Author response

Decision letter https://doi.org/10.7554/eLife.42265.037
Author response https://doi.org/10.7554/eLife.42265.038

## Additional files

### Supplementary files

• Transparent reporting form
DOI: https://doi.org/10.7554/eLife.42265.033

### Data availability

Source data have been deposited in Dyrad (doi:10.5061/dryad.5sj852c), including behavioral data, fMRI data, and lesion patient data.

The following dataset was generated:

| Author(s) | Year | Dataset title | Dataset URL | Database and Identifier |
|---|---|---|---|---|
| Wang X, Wu Q, Egan L, Gu X, Liu P, Gu H, Yang Y, Luo J, Wu Y, Gao z, Fan J | 2018 | Data from: Anterior insular cortex plays a critical role in interoceptive attention | http://dx.doi.org/10.5061/dryad.5sj852c | Dryad Digital Repository, 10.5061/dryad.5sj852c |

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
