## [Decision Letter]

[Editors’ note: a previous version of this study was rejected after peer review, but the authors submitted for reconsideration. The first decision letter after peer review is shown below.]

Thank you for submitting your work entitled "Anterior insular cortex plays a critical role in interoceptive attention" for consideration by *eLife*. Your article has been reviewed by three peer reviewers, including Klaas Enno Stephan as the Reviewing Editor and Reviewer #1, and the evaluation has been overseen by a Senior Editor. The following individuals involved in review of your submission have agreed to reveal their identity: Olivia K Faull (Reviewer #2); Micah Allen (Reviewer #3).

Our decision has been reached after consultation between the reviewers. Based on these discussions and the individual reviews below, we regret to inform you that your work will not be considered further for publication in *eLife*.

All three reviewers found the paper of interest and regarded the novel paradigm as a promising basis for developing experimental probes of respiratory interoception. However, all reviewers also expressed significant methodological concerns, particularly with regard to the absence of physiological noise correction, that prohibit an interpretation of the current findings. Since a re-analysis of the data from scratch was recommended unanimously, the decision was to reject the paper at this stage; however, we would encourage you to submit the paper again de novo, provided that the revised analyses produce results that can be considered a major advance for the field. Below, a synthesis of the three reviews is provided that is meant to help you address the methodological issues and redo the analyses. This is deliberately kept brief, with most minor comments by the reviewers excluded, to help focus on the most essential revisions.

Critical issue:

1) The fMRI data analysis did not include any correction for physiological noise in the fMRI data. Due to the breathing-related task design, it is imperative that such a correction is performed; in its absence, the current results cannot be interpreted. This is because when directing conscious attention towards breathing, it is very likely that breathing patterns will be adjusted (even when not intended). Therefore, without correction, there is no way to disentangle the effect of a change in breathing from a change in interoceptive processing of respiration-related signals. Furthermore, changes in breathing can confound fMRI measurements, both due to direct effects on BOLD signals and due to indirect effects (e.g., B0 fluctuations). For these reasons, a RETROICOR (or similar) correction would be essential for the first-level analyses. This RETROICOR correction should not only include respiratory, but also cardiac signals, given the modulation of heartrate by breathing, the representation of cardiac activity in the insula, and the presence of large blood vessels (e.g., middle cerebral artery) near the insula that can produce vascular artefacts. Additionally, we would suggest calculation and inclusion of key respiratory parameters (e.g., rate and depth during both task conditions, measures of end-tidal CO_2_ if available) as regressors of no interest in the second-level analysis, to separate physiological and interoceptive aspects of changes in breathing.

Major issues:

2) Although not quite as critical as the lack of physiological noise correction, all reviewers also noted issues with regard PPI analysis. Generally, it would be helpful to clarify what type of PPI you are using. For example, the classical PPI that tries to capture interaction effects (Friston et al., 1997); or a PPI term within an extended statistical model that tests for context-dependent coupling over and beyond any other experimental effects. If you are going for the former, the seed region would be identified by one of the main effects (please note that your F-contrast is not, as stated in the paper, statistically independent from the t-contrast of the main effect of interoceptive vs. exteroceptive attention), and the PPI term would correspond to the interaction between the other main effect and the timeseries (see Friston et al., 1997 for details). If you have in mind the latter, it would be good to ensure that the PPI model contains all experimental effects (e.g., see https://fsl.fmrib.ox.ac.uk/fsl/fslwiki/PPIFAQ).

3) The Introduction and Discussion do not represent the literature on interoception well and should be revised carefully. This includes incorrect statements (such as the assertion that breathing "is the sole 'perceptible' internal bodily signal"), incorrect citation of literature (e.g., papers cited by Craig, Critchley, etc. are not about interoceptive "attention" but about conscious awareness of/sensitivity to interoceptive signals), and lack of references to key components of the literature (e.g., theoretical papers on different components of interoception and experimental papers on insula lesions).

4) The conceptual interpretation of the experimental paradigm as an "interoceptive attention" task should be revisited. While the task relies on shifting attention (between intero-and exteroceptive domains), it primarily serves to provide a measure of intero- and exteroceptive accuracy or sensitivity (as also reflected by your analysis in terms of d'), and appears to be more adequately described in these terms.

5) The proposed experimental paradigm is innovative and has much potential for future studies of respiratory interoception. However, there are some potential problems that may need consideration. First, the control condition appears to require a different cognitive process than the condition of interest. The latter requires a temporally extended matching process; the former requires a detection process that terminates once a dot has appeared. Second, the delay was a set interval of 400 ms, rather than a proportion of the individual's respiratory cycle. This may partially determine task difficulty and performance across individuals. Third, given that the task is novel, it would be important to see more details of task performance, e.g. plots of individual accuracy rates, analysis of reaction times and signal-detection theoretic considerations (i.e., where they more biased for either interoceptive or exteroceptive conditions?). The task seems very easy compared to standard heartbeat detection tasks: were there ceiling effects (i.e., did any participants have 100% accuracy)? Finally, the task does not represent a pure probe of interoception as respiratory processes can also be tracked using exteroceptive and proprioceptive information. It thus seems likely that participants relied on a mix of interoceptive, exteroceptive, and proprioceptive information for performing the task. These issues do not invalidate the task, but they deserve a critical and frank discussion so that the reader is aware of the limitations of the paradigm.

6) There are some issues with the statistical analysis and reporting. Exact p-values, test statistics, and standardized effect sizes should be reported for all analyses. Numerous tests are reported as one-sided; this needs to be justified or replaced by two-sided tests. Non-significant results should not be presented as evidence for the absence of a difference (e.g., in the lesion analysis); this corresponds to accepting the null hypothesis and should be replaced by a corresponding Bayesian test.

[Editors’ note: what now follows is the decision letter after the authors submitted for further consideration.]

Many thanks for submitting a revised version of your manuscript to *eLife*. It has now been seen by two of the three previous reviewers. Please excuse that this consultation process took longer than usual.

We were impressed by the effort you invested in acquiring an additional dataset with concomitant measures of cardiac and respiratory activity. However, we continue to think that the statistical analysis needs to account for task-induced variations in breathing which can profoundly impact on BOLD measurements. We did read the paper (Miller and Chapman) that you attached for justification of omitting respiratory measures from the statistical model but must confess that we did not find it very insightful in relation to the current problem; in particular, equating the current issue with "Lord's paradox" (which is a rather specific case) seems misleading.

The problem in your analysis is a very generic one: including or excluding a confound regressor that is correlated to a regressor of interest in a GLM amounts to an active decision how shared variance is interpreted – or, put differently, whether one wishes to maximise sensitivity or specificity of the analysis. We think that for a study that reports the effect of a cognitive intervention for the first time, specificity is more important: the reader would like to be assured that activations attributed to the cognitive intervention are not merely driven by physiological effects. We agree that the interaction effect should be protected against task-induced breathing changes. The main effect of task, however, is not; and it is arguably of greater importance for the message of the paper.

For these reasons, we are not convinced it is a good idea to pool the two groups and report analyses without including regressors that represent physiological (respiratory) noise. We also thought that the RETROICOR analysis presented in the response letter (2nd order respiratory regressors only and no cardiac-respiratory interactions) is unusually lenient.

In our view, these problems are too substantial to proceed with in-depth peer-review. If you would like *eLife* to continue considering the paper, we would recommend that the paper (i) reports analyses from both samples separately, (ii) discusses the potential problems of interpretation in the first sample, and (iii) includes a rigorous RETROICOR correction of breathing effects for the second sample. You could boost the statistical sensitivity of the second analysis by using the FWE-corrected activations from the first analysis in order to specify a mask for reducing the search volume for FWE correction in the second analysis. In this way, you would use the higher statistical sensitivity of the first analysis in order to identify regions where the cognitive process of interest may take place and then test in the second sample, with due consideration of potentially confounding effects, whether this can be corroborated.

We are very sorry that we cannot be more positive at this stage and understand that this must be disappointing for you, given the substantial effort you have invested in the revision of this paper. We do hope, however, that the recommendation above is helpful.

[Editors’ note: what now follows is the decision letter after the authors submitted for further consideration.]

Thank you for sending your article entitled "Anterior insular cortex plays a critical role in interoceptive attention" for peer review at *eLife*. Your article has been evaluated by three peer reviewers, one of whom is a member of our Board of Reviewing Editors, and the evaluation has been overseen by Michael Frank as the Senior Editor.

Major points:

1) The wording "… checked that the AIC ROI results were not dependent on the (independent) ROI selection.…" is confusing. Presumably you wanted to say something like "… checked how much the AIC ROI results were affected by physiological noise correction.…"? In direct relation to this point, it is rather surprising to see such little effects of physiological noise correction on insula activity. Typically, physiological noise regressors (RETROICOR) do explain a substantial amount of BOLD signal in the insula. The particular statistical test you used asks whether specific contrasts are altered by the inclusion vs. exclusion of physiological noise regressors (which is fine) but is not sensitive to the question whether insular activity is affected by physiological noise at all (as implied by your wording in the subsection “ROI analysis results of the fMRI study of the second sample”). As a sanity check, it would be worth performing an additional F-test spanning all RETROICOR regressors. If this test does not show significant insula activation, it would seem wise to double-check the RETROICOR analysis, in order to make sure there are no errors.

2) Materials and methods: "The corresponding four regressors were generated by convolving the onset vectors of each trial type with a standard canonical hemodynamic response function (HRF)". Was each trial modelled as an event or a block? The methods describe each stimulus period lasting 12 seconds, which would appear more akin to a block design for the GLM?

3) The value of the mention of CO_2_ and O_2_ in this manuscript is questionable – these effects would need to be accounted for by either measuring them and regressing them out, or using an approximation such as RVT (respiratory volume per unit of time) regressors, which do not appear to be used here. Standard cardiac and respiratory waveforms and harmonics do not account for these effects. This paradigm would likely induce very slight hyperventilation when attention is directed towards monitoring breathing curves, which would result in a decrease in expired CO_2_ over the 12 second stimulus period (and the resulting washout period), which would induce a global over-estimation of the BOLD activity related to the task. RVT regressors could be included in the RETROICOR to account for this. They actually mention that there is a difference in respiratory volume between tasks in the subsection “ROI analysis results of the fMRI study of the second sample”.

4) The main effect of the task (interoceptive attention vs exteroceptive attention) is very large (Figure 2). However, it should be noted that the participants found the interoceptive task more difficult than the exteroceptive task, and thus these differences in brain activity are very likely associated with task difficulty as well as the direction of attention. This is probably worth mentioning somewhere in the Discussion?

5) Please accept our apologies – we should have noted this earlier – but the analyses presented in the lesion study suffer from a major problem. The authors are conducting non-parametric tests between participants in the interoceptive and exteroceptive condition separately, and then interpreting the presence vs lack of significance as evidence for a specific effect of AIC lesions on interoceptive attention. This, however, is an erroneous conclusion (see also https://www.nature.com/articles/nn.2886), and instead it would be necessary to demonstrate a significant group (AIC vs control) by condition (intero vs extero) interaction. It is important to perform the correct test, especially because these are the most controversial findings of the paper. It is debatable whether previous lesion research has provided any valid evidence for AIC lesions on interoceptive sensitivity and emotional awareness (in a previous review, the authors were asked to consider this work, but this appears to have been ignored).

6) There are also some concerns – and, again our apologies for not having addressed this earlier – regarding the analyses of the correlation between task measures and questionnaires (subsection “Behavioral results of the fMRI studies”). Specifically, the use of multiple one-sided tests seems questionable and would need a strong motivation, and there is a lack of multiple comparison correction. There were also questions about power: if you pool the two samples you may have reasonable power (i.e., for an effect size of|r|=0.3, N=72 would give you 75% power of detecting a significant effect at α=0.05, two-sided) for a single test, but this would diminish when taking into account multiple comparisons (with lower α as a result). It seems fair to ask that the analysis is fully transparent with regard to power, uses two-sided tests and multiple comparison correction, and tones down the interpretation of the results considerably.

7) Following directly from the previous point, one of the reviewers made the following suggestion which we would suggest you consider: "My advice is the following: take all of the correlation analyses and throw them into a giant Bayesian correlation table with appropriate priors on correlation strength, and make these supplementary analyses flagged clearly as exploratory in nature. Flag up the ones that show Bayes factors greater than 3 under a two-sided test, and in particular discuss any who are implicated in both study 1 and study 2. Refocus the paper to emphasize the novelty and importance of understanding respiratory awareness and make it clear that the link to emotion is more speculative – a fascinating area that future large-scale studies can target by using the task presented here."

---

## [Author Response]

[Editors’ note: the author responses to the first round of peer review follow.]

Critical issue:1) The fMRI data analysis did not include any correction for physiological noise in the fMRI data. Due to the breathing-related task design, it is imperative that such a correction is performed; in its absence, the current results cannot be interpreted. This is because when directing conscious attention towards breathing, it is very likely that breathing patterns will be adjusted (even when not intended). Therefore, without correction, there is no way to disentangle the effect of a change in breathing from a change in interoceptive processing of respiration-related signals. Furthermore, changes in breathing can confound fMRI measurements, both due to direct effects on BOLD signals and due to indirect effects (e.g., B0 fluctuations). For these reasons, a RETROICOR (or similar) correction would be essential for the first-level analyses. This RETROICOR correction should not only include respiratory, but also cardiac signals, given the modulation of heartrate by breathing, the representation of cardiac activity in the insula, and the presence of large blood vessels (e.g., middle cerebral artery) near the insula that can produce vascular artefacts. Additionally, we would suggest calculation and inclusion of key respiratory parameters (e.g., rate and depth during both task conditions, measures of end-tidal CO_2_ if available) as regressors of no interest in the second-level analysis, to separate physiological and interoceptive aspects of changes in breathing.

We fully agree with the reviewers that the effect of a change in breathing patterns would not be disentangled from the effect of interoceptive processing we are interested in, which might be a confound. In the following, we respond to two aspects of this comment:

First, the effect of interoceptive processing can be isolated by using the interaction effect without correcting for physiological parameters.

We agree that the possible change in breathing patterns (i.e. the respiratory effort difference) might contribute to the difference in terms of activation in the key region of interest, i.e., the AIC. Therefore, analysis to partial out the impact of the effort difference using the physiological signals might be the solution. However, based on Miller, G. A., and Chapman, J. P. (2001). Misunderstanding analysis of covariance. Journal of abnormal psychology, 110(1), 40-48., this ANCOVA method is not an appropriate solution for this scenario. That is, when the covariate and the experimental treatment are related (share variance), the regression adjustment may remove part of the treatment effect or produce a spurious treatment effect. This would be a misuse of ANCOVA, as in the so-called “Lord’s Paradox”. In our studies, the manipulation of interoception inherently involved a change in breathing effort (as evident in the difference of respiratory volume between interoceptive and exteroceptive condition, *t*(27) = 3.90, *p* = 0.001, see Author response image 1). Therefore, removing variance associated with breathing effort in BOLD signals would not only remove considerable variance in BOLD signals associated with interoceptive attention, but may also result a spurious treatment effect. A change in breathing pattern should not be viewed as a covariate but rather as a feature inherent in the task, and regressing out the physiological signals would be an inappropriate use of analysis of covariance. Therefore, although we have the physiological signals collected for the new sample, we decided to use an alternative strategy (see below) rather than follow the suggested ANCOVA method for the final report. In the response letter, we report the results after RETROICOR correction as a comparison.

Although the main effect of interoceptive attention (the contrast of interoceptive versus exteroceptive condition) is subject to this confounding and cannot be solved using ANCOVA, the interaction effect ([delay – non-delayed]_interoception_ vs. [delay – non-delayed]_exteroception_) should not be confounded by the change in breathing. This interaction reflects the brain response to the mismatch (delay versus non-delayed) under the interoceptive condition controlling for the non-specific effect (i.e., the difference in feedback stimulus under the exteroception task condition). Therefore, a positive interaction effect represents brain response to the interoceptive processing above and beyond the physical feedback difference. To further illustrate that the interaction effect of brain response is not subject to breathing effort difference, we showed the pattern of the respiratory volume under different experimental conditions (see Author response image 2). Although there was a change in the respiratory volume between interoceptive and exteroceptive task conditions (*F_(1,27)_* = 15.88, *p* < 0.001), this difference is canceled out by using the interaction effect (*F* < 1, BF = 0.025). In the original version of this manuscript, we did not explain this logic clearly or emphasize the interaction effect. In this revised manuscript, we have added a detailed description of this logic and used the interaction effect as the index of brain responses (i.e., brain activity, connectivity, and individual difference in terms of the relationship between interaction effect and interoceptive accuracy) to interoceptive processing (subsection “Image preprocessing and statistical parametric mapping”).

**Author response image 1. respfig1:** Activation maps with and without including individual heart rate and respiratory volume as covariates in the 2^nd^ GLM. (**a**) Main effect of interoceptive attention (interoceptive task vs. exteroceptive task). (**b**) Interaction between attention type and breath-curve feedback condition ([delayed – non-delayed]_interoceptive task_ – [delayed – non-delayed]_exteroceptive task_).

**Author response image 2. respfig2:** The pattern of respiratory parameters under different task conditions.

Second, the brain activation patterns are almost the same with and without correcting physiological parameters, demonstrated in a new sample of N = 28.

We are thankful to the reviewers for pointing out that change in BOLD signals can both due to direct neural activity (induced by experimental manipulation), and due to indirect effect, such as vascular response (considered to be a confounding effect). Specifically, cerebral vascular response is sensitive to circulation of CO_2_ and O_2_, and causes a change in global cerebral blood flow (CBF) and global BOLD signal. It is evident both in human and animals that the global CBF and global BOLD response influence local stimulus-induced hemodynamic response to neural activation (Cohen et al., 2002; Friston et al., 1990; Ramsay et al., 1993; Sicard et al., 2005). Typically, a larger local stimulus-induced BOLD response occurs when global BOLD was lowered, while a smaller local stimulus-induced BOLD response occurs when global BOLD was elevated. In our study, the difference on physiologic states between interoceptive and exteroceptive task conditions might cause a change in global BOLD signals, and thus confounded the effect resulting from altered neural activity (truly experimental effect) with the effect resulting merely from global hemodynamic influence. For example, the increased respiratory depth under interoceptive task condition (even when not intended) might increase cerebral O_2_ level and lowered CO_2_ level, which leads to a reduction in global CBF and global BOLD. Therefore, the AIC activation identified from the main effect of interoceptive processing (interoception task vs. exteroception task) might just represent a change in global BOLD response (i.e. lower global BOLD leads to larger local BOLD response), rather than neural activation underlying interoceptive processing. To explore the potential effect resulted from respiratory and cardiac difference, it is worth comparing activation patterns with and without correcting physiological parameters.

The reviewers offered a method to resolve this issue, which is to regress out the physiological noises by including respiratory and cardiac signals. In the original experiment, we did not record those signals. To examine the impact of those potential confounding variables, we have collected a new sample of 28 participants using the same task with both respiratory and cardiac signals recorded. According to the reviewers’ suggestions, we applied the RETROICOR correction to the phasic aspect of cardiac (8 regressors) and respiratory (2 regressors) signals to regress out potential physiological influence, generated nuisance regressors for variations in breathing rate/volume (RV) (1 regressor) and heart rate (HR) (2 regressors) which were convolved with the “respiration response function” (RV+RRF correction) and “cardiac response function” (HR+CRF correction), respectively, in addition to the motion correction (6 regressors), for the first-level GLM. We also followed reviewers’ suggestions to include the parameters of individual heart rate and breathing volume in the second-level analysis. We also conducted the data analysis without RETROICOR correction as a comparison. Our results revealed almost the same activation patterns with and without the correction of physiological signals (see Author response images 1 and 3). Specifically, the anterior insular cortex was involved in the interoceptive processing in terms of the main effect of interoception and the interaction effect.

**Author response image 3. respfig3:** Activation maps with and without physiological correction for the 1^st^ level GLM. (**a**) Main effect of interoceptive attention (interoceptive task vs. exteroceptive task). (**b**) Interaction between attention type and breath-curve feedback condition ([delayed – non-delayed]_interoceptive task_ – [delayed – non-delayed]_exteroceptive task_). (**c**) The interaction patterns of the left and right anterior insular cortex (AIC) activity without physiological correction. (**d**) The interaction patterns of the left and right AIC activity with physiological correction.

In addition, we conducted paired *t*-test on the interoception vs. exteroception and the interaction beta maps obtained without and with physiological correction (see Author response image 4). Indeed, the physiological parameters induced a global increase in BOLD signal, which was evident in a whole brain activation comparing “without physiological correction” with “with physiological correction” beta maps (Author response image 4). However, the interoceptive processing induced neural activation still hold after physiological correction (see Author response image 1 and Author response image 2). In contrast, the interaction effect is not subject to the physiological parameters induced global activation, as revealed by a “blank brain” using paired *t*-test (Author response image 4).

**Author response image 4. respfig4:** Paired *t*-test on beta maps obtained without and with physiological correction. (**a**) using interoceptive vs. exteroceptive contrast maps. (**b**) using interaction effect beta maps.

The main result of the new sample (n = 28) replicated the result of our original sample, which is that the AIC is involved in the interoceptive processing defined by the interaction effect. We have decided to combine the data from the original and the new samples (total n = 72) to enhance the statistical power. We have conducted the corresponding analyses and updated the Materials and methods and the Results. Most of the key findings from our original experiment still hold.

Major issues:2) Although not quite as critical as the lack of physiological noise correction, all reviewers also noted issues with regard PPI analysis. Generally, it would be helpful to clarify what type of PPI you are using. For example, the classical PPI that tries to capture interaction effects (Friston et al., 1997); or a PPI term within an extended statistical model that tests for context-dependent coupling over and beyond any other experimental effects. If you are going for the former, the seed region would be identified by one of the main effects (please note that your F-contrast is not, as stated in the paper, statistically independent from the t-contrast of the main effect of interoceptive vs. exteroceptive attention), and the PPI term would correspond to the interaction between the other main effect and the timeseries (see Friston et al., 1997 for details). If you have in mind the latter, it would be good to ensure that the PPI model contains all experimental effects (e.g., see https://fsl.fmrib.ox.ac.uk/fsl/fslwiki/PPIFAQ).

We thank the reviewers for making the suggestion of the correct way to conduct the PPI analysis. Actually, our PPI analyses were intended to follow the classical PPI to capture the interaction effect in terms of a psychological process and brain activity. In this sense of a 2 × 2 factorial design, the physiological signals should be extracted from one main effect while the psychological variable should be the other. However, our original selection of seed region, which was the F-contrast of all condition, was not independent of the psychological context variable, which was the interoceptive versus exteroceptive attention. Therefore, it was incorrect. Following the correction of the reviewers, we have selected the seed region of AIC based on the main effect of feedback delay (the contrast of delayed versus non-delayed condition), and used the other main effect of interoception (the contrast of interoceptive versus exteroceptive attention) as the psychological variable (subsection “Image preprocessing and statistical parametric mapping”). One way to interpret the PPI result is that the connectivity between the AIC and other brain areas is modulated by attention to interoception in contrast to exteroception. We have reconducted PPI analyses with this correct method and updated the Results. The new results showed that there was enhanced connectivity between AIC and posterior central gyrus (as well as some regions of the cognitive control network such as frontal eye field and anterior cingulate cortex) modulated by interoceptive attention, while there was increased connectivity between AIC and visual cortex (i.e., V2/3) modulated by exteroceptive attention. Additionally, we also found that the PPI between AIC and visual cortex was predictive of individual differences in interoceptive accuracy.

3) The Introduction and Discussion do not represent the literature on interoception well and should be revised carefully. This includes incorrect statements (such as the assertion that breathing "is the sole 'perceptible' internal bodily signal"), incorrect citation of literature (e.g., papers cited by Craig, Critchley, etc. are not about interoceptive "attention" but about conscious awareness of/sensitivity to interoceptive signals), and lack of references to key components of the literature (e.g., theoretical papers on different components of interoception and experimental papers on insula lesions).

We have now revised the manuscript to make the terminology accurate. We agree that the “solo perceptible” is over strong, and we revised the statement about the breath as “the most perceptible internal bodily signal” in comparison to the heart beat which is used in almost all of the tasks examining interoception (Introduction, third paragraph). Regarding the terms of conscious awareness of/sensitivity to interoceptive signals and interoceptive attention, we define interoceptive attention as the “process” and conscious awareness as the final “outcome” of the attentional processing. We have made this point more clear by defining interoceptive attention as the attentional mechanisms in interoceptive awareness (Introduction, first paragraph). The citations we have are all for interoceptive awareness. We have also made this logic more clear in the Discussion by changing the word “interoceptive attention” to “interoceptive awareness” when referring to the outcome of the attentional process for a more appropriate use.

Regarding the literature review of insula, we have added some missed references (e.g., He et al., 2009, for the empirical study for lesion of insular cortex and Garfinkel et al., 2015, for the component of interoception (Introduction).

4) The conceptual interpretation of the experimental paradigm as an "interoceptive attention" task should be revisited. While the task relies on shifting attention (between intero-and exteroceptive domains), it primarily serves to provide a measure of intero- and exteroceptive accuracy or sensitivity (as also reflected by your analysis in terms of d'), and appears to be more adequately described in these terms.

As we explained regarding the issue of the definition of interoceptive attention, attention is the process and the awareness is the outcome. In most psychological experiments, outcome is use to index the underlying processing. Here, we followed the same logic by associating interoceptive attention with interoceptive accuracy/sensitivity (which are the outcomes). We have two task conditions in the Breath Detection Task (BDT) and Dot Detection Task (DDT). Our experimental design was block design with two tasks administered in separate runs/blocks. Interoceptive attention (and corresponding performance) is the focus, as reflected by the title of this manuscript, with exteroceptive attention as the contrast baseline. In the revised manuscript, we have made the description more clearly. For example, when referring to the concept, we used “interoceptive attention”; when referring to the task, we used “BDT” and “DDT”; when referring to performance, we used “interoceptive accuracy/sensitivity”.

5) The proposed experimental paradigm is innovative and has much potential for future studies of respiratory interoception. However, there are some potential problems that may need consideration. First, the control condition appears to require a different cognitive process than the condition of interest. The latter requires a temporally extended matching process; the former requires a detection process that terminates once a dot has appeared. Second, the delay was a set interval of 400 ms, rather than a proportion of the individual's respiratory cycle. This may partially determine task difficulty and performance across individuals. Third, given that the task is novel, it would be important to see more details of task performance, e.g. plots of individual accuracy rates, analysis of reaction times and signal-detection theoretic considerations (i.e., where they more biased for either interoceptive or exteroceptive conditions?). The task seems very easy compared to standard heartbeat detection tasks: were there ceiling effects (i.e., did any participants have 100% accuracy)? Finally, the task does not represent a pure probe of interoception as respiratory processes can also be tracked using exteroceptive and proprioceptive information. It thus seems likely that participants relied on a mix of interoceptive, exteroceptive, and proprioceptive information for performing the task. These issues do not invalidate the task, but they deserve a critical and frank discussion so that the reader is aware of the limitations of the paradigm.

We would like to thank the reviewers for the comment on the innovation of our task. Here we respond to the questions and the suggestion point by point.

1) We agree with the reviewers that the exteroceptive process would stop as soon as participants detected the dot. The same strategy would also apply to the interoceptive task: the interoception would terminate if the participant detected the non-matched/matched curve to their breath. The reviewers’ concern does exist if these two processes do not terminate at the same time point. However, we randomized the time point of the appearance of the dot trial by trial, although it is not guaranteed that the average termination time points of the two tasks are the same.

2) We agree with the reviewers that manipulating the feedback delay according to each individual’s respiratory cycle is a better way in terms of controlling for task difficulty across subjects. However, we did not do so by calculating immediate respiratory cycle online due to concern about precision issues arising from the variations of the breath curves and an appropriate implication of the algorithms. Note that the 400 ms delay was determined based on a proportion (~1/10) of an average cycle of normal healthy people which is 3~4 s/cycle. However, we agree that this fixed delay could not ensure equal subjective difficulties across participants. To acknowledge this issue and for future improvement, we have added a sentence in Materials and methods (subsection “Task implementations”, second paragraph) and in Discussion (Subsection “The interoceptive task in the respiratory domain”, last paragraph).

3) We have added the box plots for individual accuracy rate, reaction time, d-prime, and beta in the Supplementary Figure 1. Participants were less accurate and slower in the BDT than in the DDT, while were more biased in the DDT than in the BDT (see subsection “Behavioral results of the fMRI study”). There was only one participant who reached 100% accuracy in the BDT and one participant reached 100% accuracy in the DDT out of 72 participants. Here we provided the plots to show the distributions of the performance, which showed no evidence of a ceiling effect.

**Author response image 5. respfig5:** 

4) We thank the reviewers for making the insightful suggestion for a critical and frank discussion about the mixed nature of interoceptive signals with exteroceptive and proprioceptive information as in our measure using the breath detection task (BDT). The interoceptive attention is toward both interoceptive and proprioceptive information, but not exteroceptive information (Gu, Hof, Friston, and Fan, 2013). In our design, we have the contrast task of dot detection task (DDT) for a measure of exteroception, so that the subtraction of BDT and DDT leaves the components of interoceptive and proprioceptive processing which is the classical definition of interoception (of bodily somatic and visceral signals or responses). We have added a sentence in Discussion to acknowledge that: “The BDT does not represent a pure probe of interoception as respiratory processes can also be tracked using exteroceptive and proprioceptive information. It thus seems likely that participants relied on a mix of interoceptive, exteroceptive, and proprioceptive information for performing the task. In our design, we have the contrast task of dot detection task (DDT) for a measure of exteroception, so that the subtraction of BDT and DDT leaves the components of interoceptive and proprioceptive processing of interoception.”

6) There are some issues with the statistical analysis and reporting. Exact p-values, test statistics, and standardized effect sizes should be reported for all analyses. Numerous tests are reported as one-sided; this needs to be justified or replaced by two-sided tests. Non-significant results should not be presented as evidence for the absence of a difference (e.g., in the lesion analysis); this corresponds to accepting the null hypothesis and should be replaced by a corresponding Bayesian test.

We thank the reviewers for pointing out those inaccuracies in analyzing and reporting of the statistics. We have now reported exact p values, test statistics, and effect sizes for all analyses. In the revised manuscript, we have all statistical tests as two-sided except for lesion study. Although the significant differences between AIC lesion patients and normal controls survived two-tailed tests, we prefer to use one-tailed tests because we have the hypothesis that lesions of a specific brain region (e.g., AIC) would induce deficits in behavioral response. We have added a sentence in Materials and methods to justify the reason using one-tailed test in the lesion study (subsection “Behavioral data analysis of the lesion study”). Regarding to the non-significant results (e.g., in behavioral analyses of the fMRI and the lesion studies), we have followed reviewers’ suggestions to conduct Bayesian tests. We have updated the corresponding Materials and methods and Results in the revised manuscript.

[Editors' note: the author responses to the re-review follow.]

We were impressed by the effort you invested in acquiring an additional dataset with concomitant measures of cardiac and respiratory activity. However, we continue to think that the statistical analysis needs to account for task-induced variations in breathing which can profoundly impact on BOLD measurements. We did read the paper (Miller and Chapman) that you attached for justification of omitting respiratory measures from the statistical model but must confess that we did not find it very insightful in relation to the current problem; in particular, equating the current issue with "Lord's paradox" (which is a rather specific case) seems misleading.The problem in your analysis is a very generic one: including or excluding a confound regressor that is correlated to a regressor of interest in a GLM amounts to an active decision how shared variance is interpreted – or, put differently, whether one wishes to maximise sensitivity or specificity of the analysis. We think that for a study that reports the effect of a cognitive intervention for the first time, specificity is more important: the reader would like to be assured that activations attributed to the cognitive intervention are not merely driven by physiological effects. We agree that the interaction effect should be protected against task-induced breathing changes. The main effect of task, however, is not; and it is arguably of greater importance for the message of the paper.For these reasons, we are not convinced it is a good idea to pool the two groups and report analyses without including regressors that represent physiological (respiratory) noise. We also thought that the RETROICOR analysis presented in the response letter (2nd order respiratory regressors only and no cardiac-respiratory interactions) is unusually lenient.In our view, these problems are too substantial to proceed with in-depth peer-review. If you would like eLife to continue considering the paper, we would recommend that the paper (i) reports analyses from both samples separately, (ii) discusses the potential problems of interpretation in the first sample, and (iii) includes a rigorous RETROICOR correction of breathing effects for the second sample. You could boost the statistical sensitivity of the second analysis by using the FWE-corrected activations from the first analysis in order to specify a mask for reducing the search volume for FWE correction in the second analysis. In this way, you would use the higher statistical sensitivity of the first analysis in order to identify regions where the cognitive process of interest may take place and then test in the second sample, with due consideration of potentially confounding effects, whether this can be corroborated.We are very sorry that we cannot be more positive at this stage and understand that this must be disappointing for you, given the substantial effort you have invested in the revision of this paper. We do hope, however, that the recommendation above is helpful.

Thank you for making constructive suggestions. We have followed your recommendations to address the issues in the data analysis and reporting. We agree that compared to sensitivity, specificity should be prioritized when exploring the effect of cognitive processing for the first time. Therefore, it is necessary to ensure that the AIC activity reported in our study is attributed to interoceptive processing, rather than driven by physiological effects. Thus, we analyzed fMRI data of the two samples separately in the following three steps:

First, we re-conducted connectivity analyses (i.e., PPI and DCM) of the first sample. You kindly pointed out that the selection of the seed region in PPI analysis should be independent of the psychological variable. In the context of a 2 × 2 factorial design, the physiological signals should be extracted from one main effect while the psychological variable should be the other. Therefore, in the new analysis we selected the seed region of AIC based on the main effect of feedback delay (the contrast of delayed versus non-delayed condition), and used the other main effect of interoception (i.e. the contrast of interoceptive versus exteroceptive attention, which was orthogonal to the main effect of feedback delay) as the psychological variable. The new results from PPI analysis and DCM showed that there was enhanced connectivity between AIC and posterior central gyrus modulated by interoceptive attention, while there was decreased connectivity between AIC and visual cortex (i.e., V2/3) modulated by exteroceptive attention. These results were consistent with what we reported in the previous versions. The sections of connectivity analyses were updated accordingly in the manuscript.

Second, we discussed the potential problems of the interpretation in the first sample due to the confounding effect of physiological activity. Specifically, the experimental manipulation in our study would inherently cause a change in respiratory patterns (i.e. amplitude and frequency) between interoceptive and exteroceptive tasks. The difference on physiological states might cause a change in global BOLD signals, which would confound with the effect of interoceptive processing we are interested in. We have added a detailed discussion of this problem in Materials and methods, and described the solution to correct for this confounding effect in this section. Following your suggestions, we regressed out physiological artifacts in the second sample by applying a rigorous RETROICOR correction of the brain activation associated with physiological activity, which included 5^th^ order respiratory and cardiac regressors, and cardiac-respiratory interactions (see details in Materials and methods).

Third, to maximize the specificity of brain response while also boosting the sensitivity, we conducted an ROI analysis in the second sample to confirm that the involvement of AIC in interoceptive processing was not subject to physiological artifacts. Specifically, we identified the ROI of the AIC from the interaction contrast in the first sample, and then extracted the signals of the ROI in the second sample after a rigorous RETROICOR correction was applied to regress out respiratory and cardiac noises. For the ROI analysis of the second sample, there was a significant interaction between attentional focus (interoceptive and exteroceptive) and feedback (with and without delay) in both left and right AIC (left: *F*(1,27) = 5.77, *p* = 0.024; right: *F*(1,27) = 5.73, *p* = 0.024; Figure 5A), which is a confirmation of what we found from the first sample. The correlation between the interaction effect of the right AIC activity and relative interoceptive accuracy was significant (Pearson *r* = 0.36, *p* = 0.03, one-tailed). In addition, the whole brain analysis of the second sample showed that there was the significant overlap in the main and the interaction effects with and without physiological noise correction (see Figure 5—figure supplement 2). The difference of the signals of the AIC between the analyses with and without physiological corrections was not significant, suggesting that the effect of the AIC was not significantly influenced by the physiological noises (see Figure 5—figure supplement 3). Altogether, these results confirm that the AIC was actively engaged in interoceptive processing. We have updated the corresponding sections in Materials and methods and Results.

[Editors' note: the author responses to the re-review follow.]

Major points:1) The wording "… checked that the AIC ROI results were not dependent on the (independent) ROI selection.…" is confusing. Presumably you wanted to say something like "… checked how much the AIC ROI results were affected by physiological noise correction.…"? In direct relation to this point, it is rather surprising to see such little effects of physiological noise correction on insula activity. Typically, physiological noise regressors (RETROICOR) do explain a substantial amount of BOLD signal in the insula. The particular statistical test you used asks whether specific contrasts are altered by the inclusion vs. exclusion of physiological noise regressors (which is fine) but is not sensitive to the question whether insular activity is affected by physiological noise at all (as implied by your wording in the subsection “ROI analysis results of the fMRI study of the second sample”). As a sanity check, it would be worth performing an additional F-test spanning all RETROICOR regressors. If this test does not show significant insula activation, it would seem wise to double-check the RETROICOR analysis, in order to make sure there are no errors.

We thank you for your conscientious suggestions regarding RETROICOR correction. Following your suggestion, we have conducted an F-test across all RETROICOR regressors. The results revealed a significant physiological impact on the insula as well as the whole brain activation (please see Author response image 6), suggesting that the RETROICOR indeed explained a substantial amount of BOLD signal under the interoceptive condition (BDT) and exteroceptive condition (DDT). However, in our last analysis, contrasting BDT versus DDT did not reveal a significant impact of physiological noise correction on insula activation. We speculate that the reason might be that standard Fourier harmonics of RETROICOR correction do not account for all effects of physiological noises, i.e., CO_2_ and O_2_ effect caused by respiratory per unit time (RVT) as you suggested in the third comment below. Therefore, we have conducted a new physiological correction by including another two nuisance regressors of respiratory volume (RV) and heart rate (HR) that convolved with the “respiration response function (RRF)” and the “cardiac response function (CRF)” respectively (Verstynen and Deshpande, 2011). This new correction revealed a reduced activation in the insula as well as the whole brain under the main effect of interoceptive attention (the contrast of BDT vs. DDT) compared to no correction (see Figure 5—figure supplement 3A), suggesting that the RV/HR regressors indeed accounted for additional physiological noise induced by the RV and HR differences between the two tasks. However, the interaction contrast ([delayed – non-delayed] _BDT_ – [delayed – non-delayed] _DDT_) was not much affected by the physiological correction, as revealed by an almost blank brain when comparing the contrast maps without and with physiological corrections (see Figure 5—figure supplement 3C), which further confirmed that the interaction contrast was not subject to the confounding effects caused by physiological changes between tasks. With this new physiological correction, we have re-conducted the ROI analyses of the second sample. The results of the ROI analysis were consistent with what we reported in the previous version of the manuscript. We have revised that misleading sentence and updated the Materials and methods and Results sections accordingly.

**Author response image 6. respfig6:** The F-test across all RETROICOR regressors. Voxelwise p < 0.001.

2) Materials and methods: "The corresponding four regressors were generated by convolving the onset vectors of each trial type with a standard canonical hemodynamic response function (HRF)". Was each trial modelled as an event or a block? The methods describe each stimulus period lasting 12 seconds, which would appear more akin to a block design for the GLM?

We apologize that the sentence describing the generation of task-related regressors was not clearly written. We modelled each trial as a mini-block with a duration of 12 seconds. We have revised this sentence to “Each trial was modelled as an epoch-related function by specifying an onset time and a duration of 12 s. The corresponding four regressors were generated by convolving the onset of each trial with the standard canonical hemodynamic response functions (HRF) with a duration of 12 s, i.e., by convolving each trial block with HRF, equivalent to a box-car function.”

3) The value of the mention of CO_2_ and O_2_ in this manuscript is questionable – these effects would need to be accounted for by either measuring them and regressing them out, or using an approximation such as RVT (respiratory volume per unit of time) regressors, which do not appear to be used here. Standard cardiac and respiratory waveforms and harmonics do not account for these effects. This paradigm would likely induce very slight hyperventilation when attention is directed towards monitoring breathing curves, which would result in a decrease in expired CO_2_ over the 12 second stimulus period (and the resulting washout period), which would induce a global over-estimation of the BOLD activity related to the task. RVT regressors could be included in the RETROICOR to account for this. They actually mention that there is a difference in respiratory volume between tasks in the subsection “ROI analysis results of the fMRI study of the second sample”.

We are very thankful for your helpful suggestions on controlling for RVT. Following your suggestion, we have re-analyzed the data of the second fMRI study by adding another two nuisance regressors of RVe and HR according to Verstynen and Deshpande, 2011. The inclusion of those additional two regressors indeed accounted for the RV and HR difference between tasks, and we found that there was an impact of physiology correction on insula activation under the main effect of interoceptive attention (the contrast of BDT vs. DDT) (please refer to our response to the first comment above for details).

4) The main effect of the task (interoceptive attention vs exteroceptive attention) is very large (Figure 2). However, it should be noted that the participants found the interoceptive task more difficult than the exteroceptive task, and thus these differences in brain activity are very likely associated with task difficulty as well as the direction of attention. This is probably worth mentioning somewhere in the Discussion?

We agree with you that the main effect of the task (interoceptive attention vs exteroceptive attention) reflected both task-specific effect (i.e. task difficulty and respiratory characteristics difference between tasks) and attention deployment. We have discussed this point in the Materials and methods section after we described the specification of the contrasts and then clarified that the interaction contrast can disentangle the effects to some extent.

5) Please accept our apologies – we should have noted this earlier – but the analyses presented in the lesion study suffer from a major problem. The authors are conducting non-parametric tests between participants in the interoceptive and exteroceptive condition separately, and then interpreting the presence vs lack of significance as evidence for a specific effect of AIC lesions on interoceptive attention. This, however, is an erroneous conclusion (see also https://www.nature.com/articles/nn.2886), and instead it would be necessary to demonstrate a significant group (AIC vs control) by condition (intero vs extero) interaction. It is important to perform the correct test, especially because these are the most controversial findings of the paper. It is debatable whether previous lesion research has provided any valid evidence for AIC lesions on interoceptive sensitivity and emotional awareness (in a previous review, the authors were asked to consider this work, but this appears to have been ignored).

We thank you for pointing out the correct way to conduct the statistical analyses of our lesion study. We have followed your suggestion and conducted a non-parametric test for the interaction effect using R (please refer to “Nonparametric Tests for the Interaction in Two-way Factorial Designs Using R”, by Jos Feys). In specific, we used the *npIntFactRep* function (from the npIntFactRep package, see https://cran.r-project.org/web/packages/npIntFactRep/npIntFactRep.pdf) that yields an aligned rank test for an interaction in the two-way mixed design with the group (normal controls, AIC lesions, and brain damage controls) as the between-subject factor and with the task (BDT and DDT) as the within-subject factor. The results showed a significant interaction between group and task on performance accuracy (*F*_(2,21)_ = 5.19, *p* = 0.015) and discrimination sensitivity (dʹ) (*F*_(2,21)_ = 4.77, *p* = 0.023). By contrast, we did not find significant interaction effect on β (*F*_(2,21)_ <1, *p* = 0.65). We then reported simple comparison effects with our original bootstrapping tests between groups under interoceptive and exteroceptive condition separately. We have updated the Materials and methods and Results sections accordingly.

We apologize that we failed to discuss the AIC lesion patient study you mentioned. We have examined the literature investigating the impact of AIC lesions on interoceptive processes and added our speculations regarding on the inconsistency to the Discussion section. Most previous lesion studies indicated the interoceptive deficits with AIC lesions (Critchley and Garfinkel, 2017; García-Cordero et al., 2016; Ibanez, Gleichgerrcht, and Manes, 2010; Ronchi et al., 2015; Starr et al., 2009; Terasawa, Kurosaki, Ibata, Moriguchi, and Umeda, 2015; Wang et al., 2014), supporting the notion that interoceptive accuracy relies on a widely distributed network with the insular cortex as a key node (Craig, 2002; Critchley and Harrison, 2013). However, the preservations of interoceptive (Khalsa, Rudrauf, Feinstein, and Tranel, 2009) and self-awareness across a large battery of tests (Philippi et al., 2012) were documented in one patient with bilateral insula damages. These studies are mostly based on subjective report focusing on the “feeling/awareness” (Khalsa et al., 2009) that might be compensated by other brain structures such as brainstem and subcortical structures, e.g., nucleus tractus solitaries, the parabrachial nucleus, area postrema and hypothalamus (A. Damasio, Damasio, and Tranel, 2012), frontal (IFG) and temporal regions, e.g., amygdala, superior temporal gyrus, and temporal pole (García-Cordero et al., 2016; Shany-Ur et al., 2014). In the current study, the BDT challenged interoceptive attention that requires the integration of interoceptive awareness and accuracy. Our examination of interoceptive attention in patients with focal AIC lesion showed that lesions of the AIC were associated with the deficit in the performance, indicating that the AIC is critical in supporting the precision of interoceptive processing.

6) There are also some concerns – and, again our apologies for not having addressed this earlier – regarding the analyses of the correlation between task measures and questionnaires (subsection “Behavioral results of the fMRI studies”). Specifically, the use of multiple one-sided tests seems questionable and would need a strong motivation, and there is a lack of multiple comparison correction. There were also questions about power: if you pool the two samples you may have reasonable power (i.e., for an effect size of|r|=0.3, N=72 would give you 75% power of detecting a significant effect at alpha=0.05, two-sided) for a single test, but this would diminish when taking into account multiple comparisons (with lower alpha as a result). It seems fair to ask that the analysis is fully transparent with regard to power, uses two-sided tests and multiple comparison correction, and tones down the interpretation of the results considerably.

We agree with you that one-sided tests without multiple comparisons correction are questionable when we do not have a strong hypothesis. In the revised version, we have added a correlational analysis by pooling the two samples (N = 72), while also reporting the results separately for each sample to keep consistent with the imaging results. For all the tests, we have used two-sided tests with multiple comparison corrections. Indeed, the correlations between interoceptive performance and questionnaire scores did not survive after the correction. Therefore, we have toned down the interpretations of interoceptive attention in relation to emotional awareness.

7) Following directly from the previous point, one of the reviewers made the following suggestion which we would suggest you consider: "My advice is the following: take all of the correlation analyses and throw them into a giant Bayesian correlation table with appropriate priors on correlation strength, and make these supplementary analyses flagged clearly as exploratory in nature. Flag up the ones that show Bayes factors greater than 3 under a two-sided test, and in particular discuss any who are implicated in both study 1 and study 2. Refocus the paper to emphasize the novelty and importance of understanding respiratory awareness and make it clear that the link to emotion is more speculative – a fascinating area that future large-scale studies can target by using the task presented here."

We appreciate your helpful suggestions. We have conducted Bayesian correlation analyses for sample 1 and 2, and also the pooled sample by specifying a beta prior. Then, we flagged up the ones showing Bayes factor greater than 3 under a two-sided test (please see Supplementary file 3). Results showed that the interoceptive performance was only robustly correlated with subjective difficulty but not with other questionnaires. Following your suggestions, we have toned down the interpretations in Discussion.